# Recent Advances in Hybrid Biomimetic Polymer-Based Films: from Assembly to Applications

**DOI:** 10.3390/polym12051003

**Published:** 2020-04-26

**Authors:** Agata Krywko-Cendrowska, Stefano di Leone, Maryame Bina, Saziye Yorulmaz-Avsar, Cornelia G. Palivan, Wolfgang Meier

**Affiliations:** Department of Chemistry, University of Basel, Mattenstrasse 24a, BPR 1096, 4058 Basel, Switzerland; a.krywko@unibas.ch (A.K.-C.); stefano.dileone@unibas.ch (S.d.L.); maryame.bina@unibas.ch (M.B.); saziye.yorulmazavsar@unibas.ch (S.Y.-A.)

**Keywords:** biomimicry, hybrid films, polymer brushes, tethered membranes, polymer cushions, amphiphilic block copolymers, biosensing, biomedical applications

## Abstract

Biological membranes, in addition to being a cell boundary, can host a variety of proteins that are involved in different biological functions, including selective nutrient transport, signal transduction, inter- and intra-cellular communication, and cell-cell recognition. Due to their extreme complexity, there has been an increasing interest in developing model membrane systems of controlled properties based on combinations of polymers and different biomacromolecules, i.e., polymer-based hybrid films. In this review, we have highlighted recent advances in the development and applications of hybrid biomimetic planar systems based on different polymeric species. We have focused in particular on hybrid films based on (i) polyelectrolytes, (ii) polymer brushes, as well as (iii) tethers and cushions formed from synthetic polymers, and (iv) block copolymers and their combinations with biomacromolecules, such as lipids, proteins, enzymes, biopolymers, and chosen nanoparticles. In this respect, multiple approaches to the synthesis, characterization, and processing of such hybrid films have been presented. The review has further exemplified their bioengineering, biomedical, and environmental applications, in dependence on the composition and properties of the respective hybrids. We believed that this comprehensive review would be of interest to both the specialists in the field of biomimicry as well as persons entering the field.

## 1. Introduction

Biological membranes play crucial roles in cellular protection as well as in the control and the transport of nutrients [1]. Biological membranes constitute an interface between cells and their surroundings, form distinct compartments within the cell [2], and provide a catalytic site for various mechanisms, such as molecular recognition, enzymatic catalysis, cellular adhesion, and membrane fusion [1,2,3]. Biological membranes consist of a double sheet of lipid molecules, generally referred to as the phospholipid bilayer. In addition to the various types of lipids that occur in biological membranes, membrane proteins and sugars are also key components of the structure [4]. While the phospholipid bilayer provides the structural backbone of the membrane, proteins, e.g., peripheral or transmembrane proteins, are incorporated in or attached to the phospholipid matrix. In the past few decades, much attention has been devoted to developing a novel model biointerfaces that mimic the basic functions of biological membranes [5]. Typically, model membranes consist of lipids (naturally occurring or synthetic) or synthetic block copolymers [6]. Native membrane lipids confer biocompatibility and biofunctionality upon the model membrane, while the synthetic block copolymers improve membrane stability, act as barriers, and allow for chemical diversity of the model membrane. In order to take advantage of the specific characteristics of both lipids and block copolymers, these two components have been mixed to form hybrid membranes [6]. Including block copolymers in hybrid membranes increases their mechanical and structural stability, which are essential features in biomedical applications [6,7,8]. Due to the importance of this topic, many reports have been published regarding hybrid membranes with a focus on the principles of assembly in solution and on supporting surfaces [9,10,11,12]. Depending on the composition and properties of the membrane, it can allow a number of diversified moieties, including polyelectrolytes [13], nanoparticles [14], biomacromolecules [15], and cells [16], to be incorporated into the system, which, due to their broad range of properties, offers functionalization potential limited only by the researcher’s imagination.

The goal of this review was to highlight some of the recent significant advances in the development of hybrid biomimetic polymer-based films holding great potential for addressing the current niches and unanswered questions from the areas of biomedical, biosensing, or environment-related applications. We have presented these hybrid films in particular regarding their applications for targeted drug delivery, stimuli-responsive coatings, tissue engineering, and biosensing by utilizing their diverse physicochemical properties, such as catalytic and optical activity, as well as stimuli-responsive changes in their morphology, chemical composition, or electrical properties. We have focused in particular on hybrid films based on (i) polyelectrolytes, (ii) polymer brushes, as well as (iii) tethers and cushions formed from synthetic polymers, and (iv) block copolymers in combination with lipids, biomolecules, and chosen nanoparticles (Figure 1).

First, we have described biomimetic hybrids based on polyelectrolytes and diverse biomacromolecules, natural polymers, and nanoparticles, followed by the presentation of solid-supported films based on polymer brushes produced either via bottom-up or top-down approach. Next, polymer-lipid tethered and cushioned composite films have been discussed with respect to their composition and properties. In the fourth part, hybrid membranes composed of block copolymers and lipids have been presented. For each system, the assembly approach has been presented, in particular layer-by-layer, bottom-up, and top-down (grafting “from” and grafting “to”, respectively), Langmuir–Blodgett and Langmuir–Schaefer methods, and some of the analytical methods commonly utilized for characterization of such films have been mentioned. The review has further exemplified their distinct properties and the properties-related applications for the development of biosensors [17,18,19,20,21,22,23,24,25,26,27,28,29,30], drug delivery platforms [16,31,32,33,34,35,36,37,38,39,40,41,42,43], cell culture platforms [44,45], biomimicry [46,47], bioelectronic devices, e.g., [48], catalytic matrices [49,50,51], biomimetic lubricants [52,53], or anti-biofouling coatings, e.g., [14,54], among others, illustrated with the most interesting chosen examples based on the recent reports. Due to the aforementioned, this work has presented, in a comprehensive manner, multiple approaches to the synthesis, processing, and applications of hybrid biomimetic films based on various polymer species, which, in our opinion, would be interesting for both the specialists in the field of biomimicry as well as researchers intending to enter the field.

## 2. Planar Hybrid Systems Based on Polyelectrolytes

Polyelectrolytes (PE) are polymers, which, when dissolved in a polar solvent like water, bear a number of charged groups covalently linked to them, and they are classified as anionic, cationic, or ampholytic, according to whether the ionized polymer carries negative, positive, or both charges, respectively [55,56,57,58,59,60,61,62,63,64,65]. Polyelectrolytes are either biological in origin, such as polynucleotides, polypeptides, and polysaccharides, or synthetic. This part revolves around hybrid films based on the synthetic polyelectrolytes.

### 2.1. Assembly of Hybrid PE-Based Membranes

The self-assembly of the films based on polyelectrolytes is typically achieved via layer-by-layer (LbL) deposition, which takes advantage of the unbalanced charge on the polyelectrolyte species while in an aqueous solution by alternating electrostatic layer-by-layer adsorption of cationic and anionic PE onto a substrate [64,65,66,67,68,69]. While the most commonly utilized approach for LbL self-assembly is the dipping procedure (Figure 2A [70]), it can be also accomplished via spin- (Figure 2B) or spray-assisted approach (Figure 2C) and sequential brushing (Figure 3) [71]. Additionally, due to their properties intermediate between a polymer and an electrolyte, PE has been assembled electrochemically via electropolymerization [19,72,73,74], electrochemically-assisted self-assembly based on intramolecular covalent crosslinking, or morphogenic approach, where an electrochemically produced species triggers the self-assembly of the PE hybrid film [75].

Less commonly utilized methods for the fabrication of PE-based layers include surface-initiated polymerization [76], chemical cross-linking [77,78,79], the sol-gel reaction of the copolymers [80,81,82], thermal hydrolysis/condensation reaction [83,84], photopolymerization [85], thermal cross-linking [86,87,88], photothermal reduction lithography [89], and drop-casting [90], either separately or in combination with LbL.

For the characterization of the PE-hybrid films, standard spectroscopic techniques are typically used for identification and analysis of organic compounds, like Fourier transform infrared spectroscopy (FTIR), nuclear magnetic resonance (NMR) [82], mass spectroscopy (MS), and Raman spectroscopy [72]. Depending on the composition and the predestined properties of the hybrid layers, the characterization can include surface-specific methods, e.g., x-ray photoelectron spectroscopy (XPS) [91] or electron-spin sensitive techniques, e.g., electron paramagnetic resonance (EPR) [92]. If the obtained layer contains a fluorophore or exhibits electroactive properties, the characterization is typically complemented by light-emission sensitive methods, such as photoluminescence (PL), fluorescence spectroscopy or absorption and or emission in UV-vis, and electroanalysis [93]. Methods commonly applied for the surface observations of PE-based hybrid layers are electron microscopy (SEM) [93], field-emission scanning electron microscopy (FESEM) [91], and atomic force microscopy (AFM) [93].

### 2.2. Properties and Applications of Biomimetic Hybrid Membranes Based on PE

Hybrid PE-based composite films contain a variety of functional compounds, either natural, such as proteins [94], enzymes [95], cells [96], or synthetic, such as dyes [97,98], temperature-sensitive polymers [99], inorganic nanoparticles (NPs) [100,101,102,103], and clay [104,105]. The PE in hybrid multilayer films acts either as active component polymeric species, e.g., facilitating the electron transfer between the substrate and the other components of the system, or as matrices for the electrostatic immobilization of the other compounds [106,107,108,109]. In polyelectrolyte multilayer films containing biomacromolecules, e.g., enzymes, peptides, or nucleic acids, the role of PE is typically limited to electrostatic support for the immobilization of the remaining charged components of the system. Such functional thin films have received great attention due to their versatility in a wide variety of applications in the area of bioelectronics and (bio)sensing [110,111,112,113], biomimetic lubrication [52,53,114], drug delivery [37], anticancer therapy [20], and anti-biofouling [14] or proton-conducting platforms [76]. For example, multilayered polyelectrolyte membranes have been used for the separation of alcohol/water mixtures under pervaporation conditions [115,116], ion separation [117], water softening and desalination under nanofiltration and reverse osmosis conditions [118,119], and for size-selective separation of molecules [120]. The application-oriented properties of the membranes can be additionally enhanced via a choice of appropriate solid support for the membrane.

Since the assembly process is mainly governed by the electrostatic attraction of the oppositely charged components, overcompensation of the surface charge after the deposition of each layer is needed to ensure the formation of the consecutive layer from the oppositely charged component. The key to the successful multilayer film formation in the presence of colloidal particles is the restabilization (minimum of particle aggregation) of the suspension after each adsorption step. The physicochemical properties of the particles, which are closely related to the suspension stability, are of main importance also in the case of nanocomposite preparations for various applications [100]. Incorporation of a metal, e.g., Au, Pt, Pd, Cu, Ni [121], semiconductor, e.g., CdS, CdSe, CdTe, TiO_2_ [74,85,122,123,124,125,126,127], or wide band-gap NPs, e.g., SiO_2_ [80,81,83,128], into PE serving as matrices attracts substantial research efforts directed to the development of hybrid materials of novel catalytic [129,130,131], electronic [132,133,134], and optoelectronic functionalities [135,136,137,138,139]. Typically, the properties and applications of the polyelectrolyte hybrid films are directly linked to the characteristic properties of their component species.

#### 2.2.1. PE-Natural Polymer Hybrid Films

Naturally occurring polymers, e.g., polysaccharides or polypeptides, have attracted considerable interest in a variety of biomedical applications due to their biocompatibility, nontoxicity, renewability, and mild processing conditions. They are degradable in vivo, either enzymatically or nonenzymatically, and can be metabolized and excreted via normal physiological pathways [140]. The conjugation of natural polymers to synthetic polymers reduces immunological reactions and toxicity that occur when only a synthetic polymer is used. Most of the natural polymers can be ionized and carry electric charges, e.g., acidic or basic side groups of amino acids in polypeptides can be positively or negatively charged depending on the pH of the supporting medium [141]. The formation of PE-biopolymer hybrids follows the standard procedure used for the synthesis of plain polyelectrolyte multilayers due to the PE characteristics of natural polymers (see Section 3.1). While the synthesis of these systems is straightforward, they have a variety of applications due to the broad range and chemical properties of natural polymers. Some of the typical applications are based on the capability to direct the adsorption of biomolecules. For instance, poly(L-lysine)-graft-poly(ethylene glycol)-poly(1,1-dimethysilacyclobutane) (PLL-*g*-PEG-PDMSB) film obtained on an oxidized polydimethylsiloxane (PDMS) surface via PEG-ylation of PDMS has resulted in a PEG/water interface of an excellent protein resistance, presenting the potential for anti-biofouling applications, like in conventional microfluidics system [14]. PLL-*g*-dextran layers obtained in the analogous procedure have greatly reduced the non-specific adsorption of proteins and served as biomimetic lubricants [52,53]. On the contrary, by utilizing the nonspecific binding properties of cellulose towards many proteins, three-dimensional structured films could be obtained for enhanced protein adsorption performance [142]. While coatings based on cellulose are typically applicable for two-dimensional flat substrates, the protein adsorption performance is severely limited. Using direct self-assembly on the hybrid poly(ethylene imine)-poly(acrylic acid) (PEI-PAA) multilayer with pre-coated porous surfaces by the dissolution and precipitation of cellulose from *N*-methylmorpholine oxide (NMMO) solution, an enhanced adsorption performance for superoxide dismutase (SOD) has been obtained [142]. The effect of tunable mechanical properties on adhesion and spreading of human chondrosarcoma cells to PLL/hyaluronic acid (HA) films has been studied, and a direct dependency on Young’s modulus has been identified [77]. While the films have been obtained by LbL followed by cross-linking with (1-ethyl-3-(3-dimethylaminopropyl)carbodiimide (EDC), Young’s modulus has been varied over 2 orders of magnitude (from 3 to 400 kPa) for wet PLL/HA films by changing the EDC concentration [52,77]. Following, alternated biodegradable layers of PLL-HA, consisting of poly(lactic-co-glycolic acid) (PLGA) obtained by dipping and spray-assisted LbL, have been tested as barriers for PLL chains that diffuse within the PLL/HA reservoirs [143]. Rapid degradation of bone marrow cells seeded on these films has been observed, while films have remained stable for a couple of days, demonstrating the potential for application as coatings with induced time-scheduled cascade biological activities. On the other hand, PLL/camphorsulfonic acid (CSA)-PLL-poly(4-styrenesulfonicacid) sodium salt (PSS) and PLL/HA multilayer films obtained by adsorbing basic fibroblastic growth factor (bFGF) or the insoluble fraction of interphotoreceptor matrix (IPM) have been applied as substrates for photoreceptor cells [144], with the number of bilayers and the type of terminating layer having a significant influence on the number of photoreceptor cells attached.

Incorporation of metal NPs into the layers has been reported to facilitate and/or enhance the electron transport within the layer and ensure the electrical contact of biomolecules deposited onto an electrode surface. Incorporation of gold nanoparticles (AuNPs) via self-assembly of methanesulfonic acid (MSA)-stabilized AuNPs and PLL onto Au electrode has led to high sensitivity to the charge of the outermost layer for the permeability of the probe ions—[Fe(CN)_6_]^3−/4−^ and [Ru(NH_3_)_6_]^3+/2+^ [145,146]. AuNPs-terminated multilayers are of high impedance and low permeability, and thus the charge transport occurs through defects and preferential paths within the multilayer. On the other hand, the cyclic voltammetry of ferricyanide (Fe(CN)_6_^3−/4−^) at film assemblies anchored with a monolayer of lipid-like 11-mercaptoundecanoic acid (MUA) and augmented with PLL/PSS bilayers and citrate-stabilized AuNPs [147] has revealed the importance of electronic coupling within PE-AuNPs films, while the decoupling has been easily achieved by disengaging AuNPs—electrode interactions.

#### 2.2.2. PE-Nucleic Acid Hybrid Layers

Due to the presence of an accumulated charge resulting from the dissociation of nucleic bases in an aqueous solution, nucleic acids can be considered as polyelectrolyte species. Many of their physical properties are strongly dependent on electrostatic interactions between the negatively charged DNA and the surrounding positively charged counterions [148]. Complexes of polycations and DNA, also known as polyplexes, deposited in the form of films onto a solid support, have been extensively studied in the past decade as potential gene delivery systems. The typical solid support utilized for the transdermal and intradermal DNA-delivery platforms is microneedle arrays (MN), minimally invasive devices that by-pass stratum corneum (SC) barrier and access the skin microcirculation, achieving systemic delivery by the transdermal route [31]. A successful delivery system should neutralize the negative charges of phosphate groups on the DNA backbone in order to avoid the electrostatic repulsion with the anionic cell surface, to compact the relatively bulky DNA structure to appropriate length scale for cellular internalization, and to protect the DNA from degradation due to extracellular and intracellular nuclease.

The stability of polyplex films based on DNA depends strongly on the characteristics of the polycations, as well as the nature of the added salt [149,150]. Binding interactions between DNA and cationic nanocarriers must be sufficiently strong to prevent nuclease-mediated degradation, yet weak enough to permit transcription [42]. The delivery of the genetic material occurs via decomposition of the polyplexes, with the mechanism depending on the properties of the PE component. For example, LbL-coating of MN arrays with charge reversal pH-sensitive polyelectrolytes, like oligo(sulfamethazine)-bpoly(ethylene glycol)-bpoly(amino urethane) triblock copolymer (OSM-*b*-PEG-*b*-PAEU), and loading them with DNA polyplexes on heparin has been applied for intradermal cutaneous delivery of DNA vaccines [37]. The electrostatic repulsion between heparin and OSM-*b*-PEG-*b*-PAEU has triggered the release of DNA vaccines, which have been effectively transfected into RAW 264.7 macrophage cells in vitro generating antigen-specific robust immune responses. An analogous approach based on DNA-loading of polyelectrolytes deposited via LbL onto MNs has been reported for pH-sensitive polydopamine (pDA) co-deposited with heparin and albumin [32]. The functionalization of polycarbonate MNs’ surface with pDa via a mussel-inspired process has served to obtain a uniform cationic charge density over the MN. Following, a pH-responsive layer composed of alternated heparin and albumin has been built up on top of the pDa, forming a stable matrix for the polyplex films based on plasmid DNA (pTarget-Ig-Aβ-Fc) and cationic polymers (Figure 4A), which have been evaluated as cutaneous DNA vaccination system for Alzheimer’s disease (Figure 4B). The system has been able to successfully induce a robust humoral immune response compared to conventional subcutaneous injection and an immediate and strong recall immune response as immunogenic memory has been elicited.

Other examples of the pH-dependent nucleic acid release include systems based on charge reversible modified PLL-dimethylmaleic anhydride (PLL-DMA) [40], ultra-pH-responsive oligo sulfamethazine conjugated poly(β-amino ester urethane) (OSM-(PEG-PAEU) immunostimulatory adjuvant poly(I:C), a synthetic double-stranded RNA [38], and photo-sensitive and pH-responsive poly(o-nitrobenzyl-methacrylate-co-methyl-methacrylate-co-poly(ethylene-glycol)-methacrylate) (PNMP) [151]. While PNMP is initially organic-soluble, on brief exposure to ultraviolet, cleavage of the *o*-nitrobenzyl groups converts the polymer to a weak polyelectrolyte (uv-PNMP) soluble in water above pH~6.5 [151]. This photo-switchable solubility has allowed for the release of vaccine-loaded polymer films carrying DNA, immune-stimulatory RNA, and biodegradable polycations into the immune-cell-rich epidermis. Films transferred into the skin have promoted the local transfection and controlled the persistence of DNA and adjuvants in the skin from days to weeks, inducing immune responses against a model HIV antigen comparable to electroporation in mice, enhanced memory T-cell generation, and elicited 140-fold higher gene expression in non-human primate skin than intradermal DNA injection.

An alternative mechanism of DNA cargo release mechanism is based on biodegradable PEs, which dissolve in physiological fluids, e.g., biodegradable polylactide (PLA) [33] and PLGA immobilized nanoparticles [34] loaded with transcutaneous plasmid DNA, which could be released in a controlled manner by the dissolution of the PE component of the film [35]. An enhancement of the nucleic acid load release has been achieved also by an applied electrical stimulus as the increase in the physical deterioration of the PE component has been observed for films deposited onto conductive MN arrays [109], e.g., hydrolytically degradable cationic poly(β-amino ester) [151]. The release mechanism based on the photocleavage of the polyelectrolyte has been reported for, e.g., cationic diblock copolymers containing PEG and *o*-nitrobenzyl moieties [42] and biotinylated photocleavable polyethylenimine (B-PC-PEI) [39] for which a triggered capture and release of nucleic acids have been effectively realized. In the latter case, the capture of plasmid DNA has been based on the ability of PEI to condense and bind it into a positively charged complexes [41]. In both cases, DNA release has occurred via light-induced disassembly of the PE part by the cleavage of the o-nitrobenzyl ester group, which after absorption of light in the UV and near-infrared range has severed to form a carboxylic acid and nitrosobenzaldehyde [152,153]. It is a general-acid-catalyzed reaction that prevails in buffered aqueous solutions. It proceeds via nitroso hydrates that limit the release rate of adenosine triphosphate (ATP) from ‘caged ATP’ at pH values ≤ 6. The approach of the incorporation of stimuli-responsive functional groups into polymer/nucleic acid complexes provides complementary functionality for both stable encapsulation and the triggered release of nucleic acids and thus is an attractive alternative to the present strategies for efficient gene delivery. Additionally, due to the same sign of the accumulated charge on DNA and RNA molecules in an aqueous solution, RNA and DNA polyplexes can be formed via an analogous mechanism [154,155] and thus making them interchangeable in the gene delivery systems described above.

Stable multilayer films have been obtained by using solely negatively charged DNA (polyAG and polyTC DNA) sequentially assembled atop PEI basing on lateral movement of the two DNA strands occurring across the homopolymeric region, i.e., slippage [128]. While the exposure to solutions of low ionic strength, the multilayers have resulted in the disassembly of the films, and they have remained stable in high ionic strength solutions. The treatment of the film with a high concentration of urea at a high ionic strength buffer has caused the reversible denaturation of the DNA, but the extensive hydrogen bonding in the film and entanglement of the oligomers have prevented complete disassembly. After the removal of urea, the DNA has renaturated, but in a more “cross-linked” manner, with hybridization of the oligomers occurring across multiple layers of the film. This reorganization of the film prompts potential applicability not only as gene delivery platforms but also in the area of sensing devices based on a switchable morphology. The interaction of polyelectrolyte and nucleic acids has been proposed in the development of sensing platforms, which are typically composed of a cationic polyelectrolyte component and a nucleotide probe. For example, a multicomponent planar system composed of poly(diallyldimethylammonium chloride) (PDDA)-PSS alternated layers, deposited onto silver nanoparticles (AgNPs)-modified glass slide, topped with PDDA, and functionalized with neutral peptide nucleic acid (PNA) has exhibited a significant enhancement of dye emission intensity, as compared to the intrinsic dye emission observed atop a conventional glass surface [91,156]. The system has been utilized for optically amplified DNA detection based on fluorescence resonance energy transfer (FRET) [23]. The role of the AgNPs has been to increase the effective field experienced by the light-harvesting PE (poly(bithiophene fluorenebenzothiadiazole), PFBT), which has been translated into higher dye signal intensities for a simple to use single-stranded DNA (ssDNA) sensing platform. The detection of DNA has been based on different electrostatic interactions and conformational structures, e.g., between cationic poly(3-alkoxy-4-methylthiophene)s and single-stranded DNAs (ssDNA) or hybridized dsDNA, which results in a change of color from yellow to red due to the complexation of ssDNA. Upon addition of an ssDNA complementary to the capture strand, triplex formation results in a return to the yellow color [157]. Other examples of cationic PE-based conjugated platforms for DNA detection include a poly{9,9-di[3-(1-ethyl-1,1-dimethyl ammonio)propyl]-2,7-fluorenyl-alt-1,4-phenylene dibromide} (PFN)–ssDNA [107], poly{(1,4-phenylene)-2,7-[9,9-bis(6‘-*N,N,N*-trimethylammonium)-hexyl fluorene]dibromide} [158], trimethylphosphonium-substituted polyfluorenes (PTMPHFs)-ssDNA [159], and poly[(9,9-bis(6′-*N,N,N*-trimethylammonium)hexyl) fluorene-alt-1,4-phenylene] bromide) (PFPBr)-AuNPs-ssDNA [160], PFBT–ssDNA [161]. On the other hand, the conjunction of PE-SiO_2_NPs and a purine base results in an enhanced electroactive sorption capacity towards purine bases present in a DNA molecule [162]. This system has been applied for electrochemical determination of adenine in adenosine triphosphate and purine bases in hydrolyzed solutions of DNA at glassy carbon electrode (GCE) surface decorated with the film. The oxidation signals have been observed in phosphate buffer solution at pH 3.5, and the current values have been proportional to their concentration in the range of 0.02–0.50 mM, with a lower detection limit (LOD) of 0.015 mM, demonstrating good metrological characteristics. In fact, due to their solubility in water, biocompatibility [163,164,165], and exceptional mechanical properties [166,167,168], SiO_2_NPs are frequently used for the fabrication of PE-composite multilayers for biomedical and sensing applications. Also, as an active species, the PE-SiO_2_ network has acted as an immobilization platform to produce other PE-based sensors, e.g., for optical sensing of tetracycline antibiotics [25].

Besides gene delivery and sensing of nucleic acid, PE-DNA film has been reported as a catalytic matrix for the synthesis of metal NPs. Due to the intrinsic affinity of DNA for transition metal ions, ionic gold has been efficiently reduced inside the DNA-based multilayer, e.g., PSSNa/poly(allylamine hydrochloride) (PAH)/DNA [36] or poly(diallyldimethylammonium chloride) (PDADMAC)-DNA [49]. In the latter case, catalytically active AuNPs, sized between 2 and 4 nm, have been uniformly distributed within the PE membrane, which, at the same time, has acted as a scaffold matrix locally immobilizing the NPs and thus preventing their aggregation. An analogous approach has been utilized for the synthesis of Ag, Pt, and Cu NPs [50,51]. It has been shown that the synthesis mechanism has not been metal-dependent, and the NPs embedded into the film have preserved their specific features, i.e., fluorescence emission in the case of copper nanoparticles (CuNPs) [50] or antimicrobial activity of AgNPs [51]. Recently, an elegant approach to combine metallic conjugated polyelectrolyte, poly(4-(2,3-dihydrothieno[3,4-b][1,4]dioxin-2-yl-methoxy)-1-butanesulfonic acid (PEDOT-S), and surfactant-functionalized DNA (DNA:cetyltrimethylammonium chloride (CTMA):PEDOT-S) into a processable electron transfer material (ETM) for photovoltaic applications has been reported [169]. This ETM has demonstrated to effectively work for bulk-heterojunction organic photovoltaic devices based on different electron acceptor materials.

#### 2.2.3. PE-Protein Hybrid Multilayers

PE multilayers can be functionalized by adsorbing proteins at the top of the architecture or by inserting one or more kinds of proteins, the same or different, at different levels in the construction during the buildup [106]. Since the applications of hybrid PE-protein membranes depend on the protein embedded within, it is necessary that it preserves its specific functionality. For this, the proteins must be able to conserve a structure close to their native one and, while being incorporated into the film, still be able to interact with species deposited at the top of the film.

Studies of the adsorption of different proteins at the surface of PSS/PAA multilayers has shown that the amounts adsorbed are greater when the overall charge of the proteins is opposite to that of the multilayer, although, even when the protein charges are of the same sign as the multilayer, a non-zero deposition has been observed **[170]**.

Numerous proteins remain biologically active after their incorporation into the multilayers. Likewise, immunoglobulin G, buried under a small number of PE layers, PSS, and PAH, has conserved its activity and interacted with its antigens [17]. As observed by AFM and SEM, either layered or disordered films have been formed depending on the number of polyelectrolyte layers separating each protein layer; when each anti-IgG layer has been separated by one PE layer, an open, disordered film structure has been observed, and significant protein aggregation has occurred. In contrast, for films in which the anti-IgG layers have been separated by five polyelectrolyte layers, a layered structure with uniform protein layers has been formed. Both types of films fabricated have presented high applicability in biosensing; the first has provided ordered, functional protein layers for sensing investigations, and the second one has served as a functional film for applications where an increased binding capacity of the film is sought. FTIR has been used to show that proteins inserted in multilayer films conserve the secondary structure of their native form [171]. In addition, the presence of the polyelectrolytes prevents the formation of intermolecular β sheets and stabilizes proteins like fibrinogen from thermal denaturation [106].

By using fluorescent proteins, it has been shown that proteins like protein A, buried within a linearly grown film like PLL/PLGA, have been unable to diffuse in the direction vertical to the film and, therefore, have remained confined within their deposition layer [172]. On the contrary, lateral diffusion has been observed for, e.g., human serum albumin (HSA) molecules adsorbed or inserted within linearly grown films like PSS/PAA [173,174], as well as for bovine serum albumin (BSA) within nanoporous PAA/PAH composite layers grown by a combination of LbL and chemical cross-linking with SiO_2_NPs [18]. In the latter case, the mechanism of molecular imprinting has been utilized where BSA molecules have been removed after the deposition to form pores ready to bind BSA from an external solution. A different approach for the deposition has been utilized in order to obtain an electroactive layer containing hemoglobin (Hb) [19]: first, *o*-phenylenediamine (PDA) has been electropolymerized onto GCE, followed by the alternating assembly of AuNPs and PDDA and topped with Hb. The multilayer has exhibited a broad linear range for electrocatalytic H_2_O_2_ reduction ranging from 1.3 µM to 1.4 mM with LOD as low as 0.8 µM, along with the rapid response, long-term stability, and without the need to dilute the analyzed solutions. On the other hand, the incorporation of an anachelin into chromophore-PE (PEG) conjugate layer has resulted in its capacity to bind Fe ions with very high binding constants [175], while exhibiting high protein resistance upon exposure to whole human serum. Depending on the properties of the chosen protein, it is possible either to mimic or to inhibit biological processes. For example, hybrids based on PEG-acrylate/methacrylate have exhibited an inhibitory effect on the formation of nanometer hydroxyapatite particles analogous to biologic mineralization due to the adsorption of the polymers to the active growth sides [176]. In the case of (phosvitin-PLL)/ chondroitin sulfate (CSA)/PLL) [114], it has been neither possible to embed phosvitin as the constitutive polyanion nor to adsorb it atop preformed films. Instead, phosvitin has triggered an instant massive film disassembly, showing its potential for protein-triggered disassembly of multilayers, which could be utilized in the drug delivery systems. On the contrary, because of the physiologically soft character of polyallylamine hydrochloride (PAAH) interactions with thylakoid membranes, interfacial self-assembly approach to fabrication of complex planar functional nanostructures from biological components and synthetic polymers could be utilized [92]. Several types of protein-based multifunctional arrays have been obtained via their simultaneous immobilization. For example, by adsorption of antigen and antibody microarrays on the ((PAAH-poly(vinylsulfonic acid) (PVS))_n_-PAAH)-coated glass, direct and indirect immunoassays multiple analyte detection has been fabricated [177]. Using this approach, a prototype of antigen microarray in a sandwich form containing alpha-1-fetoprotein (AFP), goat immunoglobulin G (IgG), and human serum albumin (HSA) and a microarray based on the antibodies against human IgG, fibronectin, HSA, avidin, and interleukin 2 (IL-2) protein have been obtained. The target detection has been realized indirectly by measuring the intensity of the streptavidin-conjugated label bound to the detection antibody with LOD as low as 1–10 pg/mL in a manner analogous to that of the widely applied enzyme-linked immunosorbent assay (ELISA). By fabrication of cytokine-based protein fluorescence microarray, simultaneous and effective detection of many cytokines has been achieved, while the capture of proteins via noncovalent adsorption on polyethylenterephthalate (PET) film has minimized the denaturation of the biological function of the proteins [178]. The system has exhibited the detection range of ca. 100-fold greater in protein array than in a typical ELISA procedure, e.g., 10 pg/mL for tumor necrosis factor α (TNF-α). The variation coefficient (CV) has ranged from 5% at 100 ng/mL to 8% at 6.4 pg/mL among the 18 spots at each concentration, as compared to CV value of 20% for ELISA, being an attractive alternative for this traditional model for cytokine detection.

#### 2.2.4. PE-Enzyme Conjugated Planar Systems

Similarly, as for protein molecules, it has been shown that glucose isomerase, glucosamylase, glucose oxidase, and peroxidase included in polyelectrolyte multilayers fully conserve their enzyme activity [106]. The applications of these multilayers are directly related to the characteristic reaction of the enzyme embedded within and can be utilized as biosensors for biomedical applications. For example, by alternated immobilization of horseradish peroxidase (HRP) and PDDA onto multi-walled carbon nanotubes (MWCNT) and its further conjugation with AFP secondary antibodies (Ab2) as the enzyme label, an ultrasensitive chemiluminescence immunoassay of an enhanced sensitivity for the detection of a cancer biomarker in human serum has been formed [20]. The sensitivity of biosensors based on a combination of PE and enzymes can be further improved by either the choice of an appropriate substrate for the formation of the membrane or by functionalization of its surface. In a study featuring a series of electroactive nanocomposite films fabricated via LbL on MWCNT, it has been demonstrated that the substrate could facilitate the electron transfer between the electrode and the enzymes, i.e., glucose oxidase (GOx), choline oxidase (ChO), acetylcholinesterase (AChE), and HRP, resulting in low LOD values for glucose, choline, organophosphate pesticide (OPP), and nerve agents (NAs), either for single-enzyme or two-enzyme arrays and the electrode [21,22]. The performance of such biosensors has been additionally enhanced by the incorporation of metal NPs, which has improved both its selectivity and sensitivity [24]. Incorporation of, e.g., AuNPs into GOx-coated metallic cotton fiber-based hybrid biofuel cell has ensured an efficient electrical contact between the anodic enzyme and the conductive support [48]. Here, AuNPs equipped with small organic linkers have been assembled onto cotton fibers, drastically improving their conductivity constituting conductive support for the GOx enzymatic anode. The resulting biofuel cell has exhibited a remarkable power density, significantly outperforming conventional biofuel cells, by promoting the charge transfer through the electrodes and proposing a strategy to improve the performance of biofuel cells. Similarly, electrodeposition of a thin hybrid layer based on polyaniline (PANI) and AuNPs capped with 2-mercaptoethane sulfonic acid onto the Au electrode has resulted in ca. 25-fold enhancement of the charge transport as compared to the analogous PANI/PSS system [146]. After a combination with GOx, the assemblies have been utilized as catalysts for the electrochemical oxidation of ascorbic acid and as electron-transfer mediators for the bioelectrocatalytic activation of glucose oxidase (GOx) toward the oxidation of glucose, proving high applicability of the system in electrocatalysis.

Some of the other applications of PE-based biohybrid systems include drug delivery [179] or incorporation of cells via encapsulation [96], single or in clusters, for potential biomedical applications like guiding cell behavior **[180]**, improved rigidity of the cells [181], or mimicking of the biomatrices [182]. While the assembly of PE-based hybrid films can be achieved in a relatively straightforward manner, the properties of these strongly depend on the properties of the incorporated biomolecule, often setting criteria for the general stability of the multilayers, i.e., thermal and time stability, storage conditions, and the need of measurements in aqueous media of near-physiological properties. Furthermore, although PE monomers in the dissolved form contain many charged groups, they are poor conductors when assembled into a film, especially when their thickness exceeds a few tens of nanometers. Recently, the research concerning the PE-based hybrid films has been revolving around their conjugation with various metal or metal oxide nanoparticles, which have been shown to significantly improve their stability and electronic properties.

## 3. Hybrid Films Based on Polymer Brushes

Polymer brushes are known as polymer films where one end of the polymer chain is attached to a solid support via covalent binding or physisorption, while the other end faces the solution or the air interface. While designing them, several aspects, such as the choice of substrate, the polymers, the biomolecules, and the type of desired interaction, between all these elements should be considered. For instance, the nature of the substrate determines the functional end groups of the polymers where they might be covalently bound. In addition to each of the building blocks, how to prepare polymer brushes on the solid support is of critical importance, and two main approaches have been applied: the bottom-up approach grafting “from” and the top-down approach grafting “to”. The grafting “from” approach allows adjusting the desired density and functionality of polymer brushes, whereas the grafting “to” approach generally leads to the formation of polymer brushes of lower homogeneity and lower surface coverage (Figure 5) [183]. Moreover, if the polymer chains are physiosorbed to the solid support, the resulting polymer brushes display lower thermal stability and robustness towards solvents compared to covalently bonded brushes. The detailed information regarding approaches on how to make polymer brushes are given below.

### 3.1. Polymer Films Obtained via the Grafting “from” Approach

The bottom-up approach (grafting “from”) refers to the in-situ polymerization from reactive species present on the substrate surface, such as immobilized initiators [184]. Depending on the initiator coupled to the substrate, various polymerization methods have been utilized, e.g., atom transfer radical polymerization (ATRP) [185], reversible addition-fragmentation chain transfer (RAFT) [186], nitroxide-mediated polymerization (NMP) [187], and photoiniferter-mediated polymerization (PIMP) [188]. Further examples of the fabrication of polymer brushes have been also reviewed elsewhere [189,190,191,192,193].

Currently, ATRP is the most extensively used polymerization method for grafting polymers from solid supports to form polymer brushes since it can be tailored depending on the reversible activation-deactivation equilibrium between a metal complex and a polymer chain-end in the presence of the monomer. By using this method, poly-phenoxyethylmethacrylate (PHEMA), poly(hydroxypropylmethacrylate) (PHPMA), and poly(carboxybetaine acrylamide) (PCBAA) brushes of different lengths have been formed on glass substrates [194]. PHEMA and PHPMA brushes have maintained their stability as they have shown no decomposition or decrement in film thickness while being immersed into phosphate buffer saline for up to 20 days [195], making them excellent materials for biomedical applications (Figure 6). PCBAA, however, has demonstrated a significant decrease in thickness, probably due to its high hydrosolubility compared to PHEMA and PHPMA, which means the more hydrophobic the polymer is, the higher stability the brush displays.

Moreover, the thickness of PHEMA films has proved to play a role in the stability of the cell-line, where higher film thicknesses provide a better cell environment [196]. PHEMA brushes have also displayed a strong protein resistant behavior: protein adsorption initially decreases with growing brush thickness and reaches a plateau above polymer brush thickness of 20 nm, followed by an increase in brushes with a thickness of 40–50 nm [195]. Although protein adsorption has not been promoted on PHEMA brushes, β-glucosidase enzymes have been adsorbed onto PHEMA and have maintained their orientation, conformation, and enzymatic activity [197]. The reason why β-glucosidase enzymes have functioned on PHEMA brushes would be due to noticeable changes in hydration, which might support the protein adsorption. ATRP has also been used to grow poly (methylmethacrylate) (PMMA) brushes from silicon wafer [198]. Furthermore, reverse ATRP has been employed to grow poly(styrene)-*b*-poly(methyl methacrylate) (I-PS-*b*-PMMA) block copolymer brushes in situ using Cu (II)/ligand complex [189]. This modified protocol has been based on the immobilization of an azo-initiator, which decomposes into radicals and gaseous N_2_ as heat increases due to the unstable RN=NR’ bond. However, the major drawbacks of surface-initiated ATRP remain the strict air-free conditions, the need to use harmful metal catalysts, and the reaction solutions being not recyclable [199], which limit employing ATRP for polymer brushes on the industrial scale.

Surface-initiated RAFT is an alternative method to create polymer brushes. It is compatible with a wide range of monomers, and instead of using toxic metal catalysts, RAFT uses thiocarbonylthio compounds [200,201]. This polymerization method is based on a reversible regenerative chain transfer mechanism and allows to synthesize polymers with complex molecular architectures, low polydispersity, and specific functional end groups [202]. For example, in order to form polymer brushes, azo-initiators are first immobilized onto the substrate, and then the monomers are subjected to polymerization in the presence of chain transfer agents. This approach has been employed to polymerize a variety of monomers, such as methyl methacrylate (MMA)[186], styrene [186], vinylbenzyl-trimethylammonium chloride (VBTAC) [203], glucosylureaethyl methacrylate (2-deoxy-2-N-(2′-methacryloyloxyethyl) aminocarbamyl d-glucose (GUMA) [204], and 2-(dimethylamino) ethyl methacrylate (DMAEMA) [205]. More specifically, carboxyl functionalized trithiocarbonate or dithioester RAFT agents have been used for polymerization of MMA, providing carboxyl end to MMA brushes [206,207]. This has led to easy conjugation of peptides, proteins, or carbohydrates with either alcohol or amine groups. PolyGUMA brushes have not only been resistant to nonspecific adsorption of albumin and lysozyme but also have suppressed cell adhesion (e.g., human liver cancer cell line HEPG2 and human embryonic kidney 293 cells, HEK293) due to the presence of glucosylurea groups [204]. Surface initiated RAFT [208,209] permits the production of polymer brushes of thicknesses lower than 30 nm. Despite the obvious advantages of the polymer brushes formed by RAFT, this method is relatively costly due to expensive RAFT agents and requires a multiple-step synthesis protocol. RAFT polymerization has also been used elsewhere to achieve different enzyme coupling [210,211].

NMP involves the reversible growth of the radical chain ends with stable nitroxide-free radical [212]. The main advantages of NMP are that neither specific requirements towards metal and metal complexes are needed and nor additional sophisticated purification steps. This polymerization method provides colorless and odorless polymers [213]. The first use of NMP reports 2,2,6,6-tetramethyl-piperidinyloxy (TEMPO)-based initiators [214]. However, the TEMPO mediated NMP reactions have been limited to only a few types of monomers, mainly styrene and its derivatives, and could not be applied for acrylic monomers. In this respect, attempts to incorporate nitroxides have been made [187]. For example, poly(n-butyl acrylate) (PBA) brushes have been formed on silicon substrates by using an immobilized azo-initiator and the nitroxide, *N*-tert-butyl-*N*-1-diethylphosphono-2,2-dimenthylpropylnitroxyl (DEPN) [215]. By employing an immobilized TEMPO-based alkoxyamine initiator in the presence of free 2,2,5-trimethyl-4-phenyl-3-azahexane-3-oxyl (TIPNO)-based alkoxyamine, poly(tert-butyl acrylate) (PtBA) brushes have been grown on silicon substrates [216]. The thickness of PtBA brushes has ranged between 100 and 200 nm and has been tunable depending on the molar ratio of tert-butyl acrylate and alkoxyamine. Although the surface-initiated NMP provides some unique advantages, such as wide range of polymerizable monomers, ease of use, and metal-free process, it suffers from (i) slow polymerization kinetics that involve elevated temperatures and long polymerization time, (ii) the inability to easily control the polymerization of methacrylate monomers due to side reactions, and (iii) slow recombination of the polymer radical with nitroxide [217].

In surface-initiated PIMP, compounds acting simultaneously as initiators, transfer agents, and **ter**minators, named as iniferters, are typically immobilized on a substrate. They initiate radical polymerization and generate chain transfer agents, thus leading to a living polymerization [218]. For example, organosilane-terminated iniferters have been used in the context of polymerization of styrene and MMA into homopolymer (PS or PMMA) and block copolymer brushes (PS-*b*-PMMA) [188], as well as for polymerization of DMAEMA, spiropyran-containing methacrylate (SPMA), or (*N*-isopropylacrylamide) (NIPAAM) [219]. Up to date, the maximal thickness of polymer brushes formed by PIMP has reached 800 nm [220]. In general, the PIMP method is based on a metal-free redox catalysis approach, which renders it attractive for a number of applications in biomedicine and electronics. Compared to other controlled radical polymerization methods, light-mediated polymerization offers the possibility to fabricate brushes with complex 3D microstructures. Photopolymerization has proved to be an elegant approach for in situ films’ polymerization and surface modification. Among its advantages, Spatio-temporal control of polymerization is possible, and the reaction can proceed at room temperature, thus providing economic and environmental benefits. Moreover, photopolymerization is achieved on the manufacturing scale with a wide range of monomers, which makes it a competitive process to address industrial challenges [221,222,223]. For example, a bacteria-like hybrid has been fabricated using a photopolymerization process. Glass surfaces have been functionalized using a photoinitiator immobilized at the glass interface, leading to covalently bonded pNIPAAM [224]. Furthermore, the system implements silver ions, which are reduced to antibacterial metallic silver nanoparticles when the photoinitiator decomposes into radicals upon light irradiation. Therefore, both NIPAAM polymerization to pNIPAAM and silver nanoparticles precipitation have been achieved photochemically, making the 2 in 1 system suitable for biofouling applications (Figure 7). pNIPAAM coatings incorporating AgNPs (Figure 7D) have shown significantly less bacteria adhesion than glass alone (Figure 7B) and glass-coated pNIPAAM (Figure 7C). Moreover, after immersion in a saline bath, both pNIPAAM and pNIPAAM/AgNPs films have demonstrated almost no remaining bacteria attachment (Figure 7C’,D’) compared to a bare glass substrate (Figure 7B’). This implies that bacteria can sense and avoid harmful pNIPAAM/AgNPs surfaces.

Other examples using photopolymerization to combine brushes with biomolecules have been reported elsewhere [221,222,225]. However, as the photochemical reaction needs to be conducted under light exposure, this method may not be favorable for opaque microfluidic platforms unless they are made of optically transparent materials.

### 3.2. Polymer Films Obtained via the Grafting “to” Approach

The top-down (or grafting “to”) approach involves the synthesis of polymers, followed by their attachment to the solid support [226]. This process either comprises covalent bonding or non-covalent interactions between the macromolecules and the solid support. In order to achieve the covalent attachment of the polymer chain to the solid support in a grafting “to” design, chemical modification of the polymer and/or the surface is required.

The two most commonly used substrates are gold, for which the thiol chemistry can be employed to covalently attach the polymer to the surface [227,228] and silicon oxide, the bonding to which occurs via silane chemistry [229]. A one-step grafting procedure is employed in cases where thiol or silane chemistry is compatible with both the polymer and the substrate. For example, polymer brushes based on Si(OH)_3_ [230] or triethoxy silane-terminated PS in its soft form, i.e., at temperatures above the glass transition, have been prepared on silicon surfaces [231]. Resulting properties of the polymer brushes have been found to be dependent on the temperature and time of the grafting process, initial polymer film thickness, and its molecular weight. When the grafting time is shorter, the polymer brushes are deposited in the form of inhomogeneous structures on the solid support, whereas by increasing the grafting time, homogenous polymer brushes are formed. Thiol-functionalized PS [232], polyethylene glycol (PEO) [233], pNIPAM [234], poly(*N*,*N*-dimethylaminoethylmethacrylate) (PDMAEMA) [235] have been attached by the grafting “to” method to gold to form brushes.

Apart from the thiol and silane chemistry, different chemical modifications have been reported, for example, based on the reactions between (i) epoxy and carboxy groups [236], (ii) epoxy and amino groups [237], and (iii) carboxy and amino groups [238]. Systems based on reactions between epoxy and thiol groups [239], epoxy and maleic anhydride [240], as well as click chemistry [241], have been also used, even though they are less common. Since polymers are synthesized before their attachment to the substrate, the thickness of polymer brushes prepared by grafting “to” method can be controlled by the molecular weight. In fact, higher polymer chain length leads to thicker polymer brushes. The uniformity of the polymer brushes is controlled by using polymers with a narrow polydispersity index [242]. Both polymer thickness and homogeneity are important for protein insertion; otherwise, their stability and activity might be altered. Nonetheless, due to the steric hindrance introduced by the grafting “to” process, low-density polymer films are obtained. In order to increase the grafting density, the polymer has to be attached to a solid support when in a straight coil conformation, which is not thermodynamically favorable. In this respect, techniques, such as Langmuir–Blodgett (LB) and Langmuir–Schaeffer (LS), have been used to physically attach polymers to solid supports [243,244]. LB and LS depositions simply consist of transferring a monolayer composed of polymer chains from the air–water interface onto solid support via vertical dipping (Figure 8A). Before the transfer, the monolayers at the air–water interface need to be characterized by recording the Langmuir isotherm [1,245] to obtain intrinsic information, such as the surface pressure values for the phase transitions and the breaking point of the monolayer. A phase transition occurs due to the surface pressure applied to the monolayer and takes place when the molecules change their packing from a fluid phase (liquid expanded, LE) to a more rigid gel phase (liquid condensed, LC), typically monitored in real-time using Brewster angle microscopy (BAM) [6,246,247].

LB deposition offers the advantages of direct transfer of free-standing polymer chains from the air–water interface to solid support while preventing their disruption. Plus, with LB and LS methods, both the grafting density and the chain length are controlled, leading to form highly ordered and homogenous films [249]. However, an extreme level of cleanliness is needed for a successful defect-free polymer film deposition, together with the solubility and stability of the species in chloroform. These requirements make the use of LB and LS methods limited for industrial applications [6,245,250,251]. For example, a bilayer made of amphiphilic sulfur-functionalized poly(butadiene)-*b*-poly(ethylene oxide) has been successfully attached to ultra-smooth gold upon LB and LS transfer to form homogenous polymer films [228].

### 3.3. Properties and Applications of Hybrid Films Based on Polymer Brushes

Due to their versatility, polymer brushes, coupled with biomolecules, find numerous applications, especially in the biomedical field for the development of biosensors, biofouling surfaces, scaffolds for tissue engineering, as well as for cellular studies [252,253,254,255,256,257]. More rarely, they are found in the field of microelectronics and energy harvesting from biomolecules [256,257].

The substrate nature has been shown to affect the properties of polymer brushes. For example, when grafted on indium tin oxide (ITO), the grafting density and thickness of PMMA and pNIPAM brushes have been enhanced compared to those grown on glass and silicon substrates [258]. Moreover, the kinetics of polymerization reactions using metal complexes are generally increased when the polymerization occurs from metallic substrates as compared to non-metallic ones [259]. Polymer brushes have also been studied on textured substrates, such as ordered porous anodic aluminum oxide (AAO) [260,261,262]. In this respect, the thermoresponsiveness of pNIPAM has been used to study the release of a fluorescent model molecule, calcein, entrapped in AAO upon temperature change. Such a system may find applications in lab-on-a-chip technologies, where spatially confined and thermally controlled delivery of substance is required [260]. On the other hand, the stability and durability of polymer brushes grown from various substrates have been shown to depend on the number of bonds between the initiator and the solid substrate [263]. Therefore, gold substrates have been used to produce well-defined and homogenous polymer brushes to achieve interface for molecular recognition [263,264].

Polymer brushes have been employed as biosensors since their surfaces are capable of sensing the surrounding medium following a change in light, temperature, salt concentration, and pH [265]. Upon changes in the environment, polymer brushes go through the reorganization of internal or external structures on a solid support, leading to macroscopic responses, which might be monitored by sensors. For example, electrochemical DNA sensors measure changes in electron transfer dynamics when the structural rearrangement of polymer brushes are induced by target hybridization [266]. Similar approaches are also applied for aptamer-based sensors where binding-induced aptamer folding is detected [267,268] and for polymers coupled with DNA moieties to improve the sensitivity of DNA probes [269,270].

A system incorporating dextran polymer combined with biotins proteins has been described and is termed as polymeric enzyme detection (PED). It uses a biotin-functionalized dextran derivative to increase the number of detecting species, thus lowering the detection limit of the DNA probe [271] (Figure 9A,B). This dextran-biotin hybrid has been designed to detect several genes, such as *mec*A, which is present in multiple bacteria, thus making it applicable to a wide range of microorganisms. The detection probe is coupled with 3,3’,5,5’-tetramethylbenzidine (TMB), which is a chromogenic substrate undergoing color change upon oxidation by hydrogen peroxide produced by, e.g., HRP. Spectrophotometry results have shown that the PED-enhanced detector has improved the lower limit of sensitivity 10 to 25-fold compared to the standard detection approach (Figure 9C).

In the field of immunosensing, the fabrication of planar and porous hybrid polymer-immunoglobulin devices through electro-polymerization of polyamidoamine (PAMAM) dendrimers [272] and PANI [273] has been reported. The resulting systems have shown their potential to be used as highly sensitive immunosensors for antigen capture. The response sensitivity of the porous system is 2 times higher than for the planar system, with a detection limit of 3.7 fg/mL for AFP, the expression of which is enhanced for patients having liver cancer pathology (Figure 10). The fabrication of poly(n-butyl methacrylate) (PBMA) or poly(n-butyl acrylate) (PBA) brushes on glass substrates has been reported to monitor the temperature-controlled protein adsorption [274]. Strong temperature dependence of protein adsorption rate has been demonstrated due to the increase in the number of accessible active binding sites while the polymer becomes softer. The adsorption of BSA and a specific antibody (anti-IgG) to PBMA brushes has been studied and has shown a strong temperature dependence. The adsorption rate increases at higher temperatures [274], prompting a higher number of accessible active binding sites while the polymer becomes softer, which can be utilized for cell culture-related applications.

In the domain of cell adhesion, implantable biocompatible cellular polypeptides coatings have been studied as potential systems to enhance and accelerate anchorage between an implant and the surrounding living tissues, leading to use in the implantable devices. The human osteoblast-like cell line has been used, and the system has proved its efficiency at promoting the adhesion of osteoblast cells [275]. On the other hand, photopatterned brush surfaces implementing a photoiniferter coupled to cell-adhesive arginylglycylaspartic acid (RGD)-ligands have proved their efficiency at spatially controlling cell adhesion over previously described systems [276].

Lubrication of polymer brushes is critical for artificial joint fabrications since it allows either to reduce or to control the friction forces between surfaces in sliding movements [277,278]. For example, polyionic brushes have displayed high surface lubrication properties, resulting in low friction surfaces [279], while PS polymer brushes tested under different solvent conditions have shown surface friction properties highly dependent on the environmental conditions, e.g., humidity [280]. For medical implants applications, the surface of the material has to fulfill the requirements of being antifouling upon contact with biological fluids, such as a serum, plasma, or saliva [190,281], and to minimize nonspecific adsorption of biological substances, such as proteins, cells, and bacteria, to the implant surface. Depending on the composition and morphology, polymer brushes exhibit a different degree of antifouling, antibacterial, and biocompatible properties. On a system combining zwitterionic polysulfobetaine brushes with cellulose membranes, investigation of platelet adhesion, hemolysis rate, and plasma protein adsorption onto polysulfobetaine brushes have revealed that the cellulose membrane functionalized with brushes has exhibited enhanced hemocompatibility as compared to bare cellulose membranes [210]. Biocompatible polymer brushes of hydrophilic, neutral, or zwitterionic nature have been reported, such as poly(oligo (ethylene glycol) methacrylate) (POEGMA) [192], PHEMA [282], poly (2-methacryloyloxyethyl phosphorylcholine) (PMPC) [283], poly(sulfobetaine methacrylate) (PSBMA) [284] on versatile substrates, such as silicon, gold, and polymeric surfaces. For example, POEGMA brushes, with their low fouling characteristics towards fibronectin, BSA, and lysozyme, have been reported as protein-resistant brushes for screening biological interactions where nonspecific binding must be suppressed [285].

On the other hand, protein fouling properties have been achieved by tuning polymer coatings’ thicknesses: PHEMA brushes of approximately 20 to 45 nm in thickness have displayed almost zero (<0.3 ng/cm^2^) protein adsorption, whereas brushes of thicknesses ranging from 20 to 30 nm have led to protein adsorption of <3.5 ng/cm^2^ from diluted human blood serum and plasma [195]. Sometimes specific anti-biofouling properties of polymer coatings are required towards bacteria while being biofouling for specific biomolecules. For example, GOx has been strongly immobilized on the polymer-coated surface while retaining its catalytic activity, whereas the adhesion of bacteria (*Staphylococcus epidermidis* and *Escherichia coli*) has been prevented [12].

Anti-biofouling activity towards marine organisms has been reported by using non-toxic block copolymers as an alternative to biofouling paints containing toxic metals [251]. Solid-supported polymeric films have also been developed for depollution applications, such as CO_2_ capture from flue gas [286] or flammable gases [287]. Polymer brushes have also been used for energy production by harvesting both electrochemical and mechanical energy from biomolecules [257].

## 4. Polymer-Lipid Tethered and Cushioned Composite Films

The simplest model platform is to mimic biomembrane on solid support known as solid-supported lipid bilayer [44]. If the solid support is hydrophilic enough, a thin water layer with a thickness of 1 to 2.5 nm is formed [288], which acts as a lubrication layer or reservoir between the membrane and the solid support, not only inducing similar thermodynamics properties with the free-standing membrane but also a different range of lateral mobility of the membranes [289,290]. Nevertheless, the space between the membrane and the solid support has numerous negative effects on membrane quality. For example, the membrane properties change depending on the substrates, which increases the density of defects, leads to poor lateral mobility, poor electrical sealing, and difficulty in incorporating membrane proteins [291,292]. In order to overcome these negative outcomes, the polymer is introduced between the solid support and lipid membrane as an aqueous lubricant layer, which allows the design of different polymer-lipid composite films, such as polymer tethered and polymer cushioned lipid bilayer membranes.

Moreover, the solid support is associated with the optical, electrical, or acoustic transducer to monitor membrane properties (e.g., lateral mobility, sealing, and mass, respectively). Depending on the nature of solid support, different characterization methods are applied. For example, gold is one of the most efficient substrates for surface plasmon resonance (SPR), an optical technique allowing for direct, real-time kinetic measurements of both membrane formation and membrane-protein interactions for surface and interfacial science studies. Electro-impedance spectroscopy (EIS) and cyclic voltammetry measurements are performed on conductive substrates (e.g., gold or silver), allowing the characterization of electrical properties of the membranes (e.g., conductance, resistance, and capacitance). Besides, fluorescence microscopy and fluorescence recovery after photo-bleaching (FRAP) are performed on transparent substrates (e.g., glass) to visualize different morphologies and lateral mobility of the membrane. Different materials, such as silicon oxide (SiO_2_), titanium oxide (TiO_2_), aluminum oxide (Al_2_O_3_), and ITO, have been commonly used as a solid support in their biocompatibility and numerous biomedical applications, e.g., as components of medical implants [293,294] or for biosensing applications [295].

### 4.1. Polymer-Tethered Lipid Bilayer Membranes

Polymer-tethered lipid bilayer membranes are based on supramolecular assembly of architectural elements, including solid support, polymeric tethers, and fluid lipid bilayer decorated if needed by versatile biomolecules [296]. Once the polymeric tethers are covalently attached to a solid support, the fluid lipid bilayer architecture is completed to have polymer-tethered lipid bilayer membrane either by transfer of the lipid monolayers via LB or LS techniques or vesicle fusion (Figure 8B in Section 3.2). Vesicle fusion method has been commonly utilized for single-component block copolymer- and phospholipid-based planar membranes and they could be very well employed for the polymer-lipid composite films. For example, different kinds of substrates—silica, gold, glass, or mica—have been used to prepare supported membranes due to their hydrophobicity/hydrophilicity properties [297,298,299]. Via spreading of a colloidal solution onto solid support where polymeric tethers are attached at a determined flow rate and under specific conditions (e.g., increasing the concentration of vesicle or injecting an electrolytes solution), the vesicles are induced to rupture, resulting into a supported membrane [299].

More specifically, polymeric tethers are intermediate components covalently attached to both lipid and solid support. Principally, the choice of the substrate determines the characterization method but also the polymer tether system since it is covalently attached to the substrate. Intermediary binding moieties are required for a covalent attachment between the solid support and polymer tethers. For example, alkylsilanes are used as intermediary molecules for silica and mica, whereas alkymercaptanes are used for gold and gallium arsenide (GaAs) [300,301]. Furthermore, there are a number of well-known intermediary groups that bind to specific metals, metal oxides, and semiconductors, such as thiols, sulfides, and disulfides on Au, Ag, Hg, or Pd [302], chlorosilane and alkoxysilane on silica [303,304], phosphonates on alumina and metal oxides [305], with highly diverse combinations still being developed according to the intended biotechnological applications. For example, the self-assembled monolayers (SAMs) containing sulfur-based lipoic acid moiety can provide an exceptional packing density as well as good membrane sealing of the formed monolayer, but it has not been possible to incorporate the larger proteins [306]. Additionally, changing the intermediary moiety from a single thiol to a disulfide has induced an increase of the electrical resistance of membrane from 15 MΩ cm^2^ to 55 MΩ cm^2^ [307]. In addition to sulfur-based chemistry, silane-based functionalization has been often utilized to obtain SAMs, which can be easily integrated into the standard microelectronic fabrication technology. However, resulting membranes have not offered higher membrane resistance properties compared to SAMs fabricated via sulfur-based anchor moiety. After the polymers are tethered to the solid support, the lipid bilayer is completed by the physisorption of lipid molecules. For example, a lipid bilayer composed of dimyristoylphosphatidylcholin (DMPC) and *N*-succinimidomyristidic ester has been formed onto the methoxysilane-functionalized glass with a maleic acid-based copolymer by LB followed by LS transfer [308]. *N*-succinimidomyristidic ester functions as a linker that incorporates itself/is incorporated into the lipid bilayer but also covalently bound to the polymeric support, effectively attaching the bilayer to support.

Moreover, coupling lipid bilayer to the polymer has been also achieved by the use of lipopolymers [309,310], which have a lipid structure on the end of a polymer chain, leading to its insertion into the lipid bilayer, and the other end is covalently attached to the solid support to form the complete membrane (Figure 11A). For example, the covalent attachment between poly(ethyl oxazoline) (PEOX) and glass substrate has been achieved by benzophenone silane photocoupling agent, inducing a photo cross-linking reaction at 340 nm due to n, π* transition in the carbonyl group (Figure 11B) [311]. The DMPC lipids with different ratios of lipopolymer, dioctadecylamine poly(ethyl-oxazoline) (DODA-E85) (e.g., 5%, 20%, and 30%) have been transferred to benzophenone silane functionalized substrate by Langmuir Blodgett at the surface pressure of 25 mN/m [312]. By increasing the concentration of DODA-E85 tethering density, lateral mobility of the bilayer has decreased, indicating that the hydrophobic lipopolymer moiety has functioned as an immobile obstacle and reduced the diffusion of lipids (Figure 11C). Additionally, in the absence of covalent attachment points between polymer-substrate and polymer-lipid bilayer, the polymers have been transferred to the upper leaflet of the membrane, inducing no measurable space between the lipid membrane and solid support [313].

Inspired by lipopolymers, new tethered membrane systems, which are based on lipoglycopolymers, have been also developed [315]. They are derived from naturally occurring glycocalix surrounding the cell with carbohydrates, providing a high osmotic pressure to keep cells apart from each other. The amphiphilic lipoglycopolymers that can be used for tethered membrane systems have been produced by nitroxide-mediated living free-radical polymerization [315]. This approach has allowed the placement of the dioctadecylamine (DODA) at the end or along the glycopolymer backbone. Simply, the lipoglycopolymer is composed of DODA lipid and polyacrylate backbone with β-O-3 linked glucose. Although resulting lipoglycopolymers have a narrow molecular weight distribution, small differences in length of glycopolymer tethers and substrate surface lead to homogenous covalent attachment of glycopolymer tethers, and thus there is no formation of planar membrane architecture. In order to solve this problem, lipoglycopolymers have been oriented at the air–water interface and transferred to substrate via LB, and then they have been covalently attached to the substrate via an azide-based photochemical reaction. Highly insulating lipoglycopolymers-based tethered membranes with large separation distances of more than 10  nm from the surface have been produced with impedance varying from 1 to 3 MΩ cm^2^ [316].

### 4.2. Polymer-Cushioned Lipid Bilayer Membranes

When a soft hydrated polymer is used as a cushion between the substrate and the membrane, it should ideally serve as a lubricant and enable it to avoid the direct contact of the membrane from the substrate. Besides, polymer cushion mimics the cytoskeleton of cells if it is properly selected [314]. The sub-membrane space depends on the polydispersity, nature, and density of the polymer, as well as its swelling behavior upon hydration. Typically, polymer cushions are attached to solid supports via physical adsorption. The separation between the membrane and substrate relies on electrostatic interactions. Polymer cushioned platforms have poor stability when the attractive electrostatic interactions between the cushion and the lipid membrane are relatively weak, leading to the polymer cushion being easily detached from the surface. On the other hand, if the attractive electrostatic interactions between the layer and the polymer cushion are too strong, it might reduce the lateral mobility of the membrane due to proximity effects. Moreover, since it is difficult to control the morphology, swelling behavior, and thickness of the polymer cushion, the electrical sealing properties of lipid membranes are usually not sufficient for electrochemical studies, making them only amenable to study membrane-associated processes that do not involve any molecular transport events through the membrane. Nevertheless, there have been successful examples on polymer cushioned membranes, such as hydrogels [312], cellulose or hairy rod polymer, chitosan [317,318,319], pH-responsive poly[2-(dimethylamino)ethyl methacrylate-block-methyl methacrylate] (PDMMA) diblock copolymer [320], PEI [321], polyelectrolytes [322], and PEG [323]. Although there is no covalent interaction between substrate and polymer cushion, phospholipids have been covalently attached to polymer cushion in some cases [324,325,326].

For example, 8–9 nm thick maleic acid-based copolymer has been attached to the methoxysilane-functionalized glass and used as a hydrogel cushion membranes after deposition of DMPC (dimyristoylphosphatidylcholin) and *N*-succinimidomyristidic ester via LB, followed by LS transfer [312]. Lateral mobility of lipids in a pure DMPC bilayer formed on an un-functionalized glass has reached the speed of 4 μm^2^/s, while functionalization of the glass with hydrogel cushion has decreased the lateral mobility to 3 μm^2^/s. In the case of hairy rod polymer or cellulose-cushioned membrane platforms, the LB technique is typically employed to deposit the multilayered polymer cushion onto the alkyl functionalized substrate, followed by the fusion of vesicles onto the surface of the substrate coated by the polymer molecules. For example, the functional incorporation of integrin into the cellulose-based cushioned membrane has been monitored [327]. Although the preparation of the cellulose-cushioned membrane is not straightforward, it has been possible to control both the density and the mobility of incorporated integrin receptors by reducing the adhesion forces between substrate and integrin. Since PDMMA and PEI-based polymer cushions are mostly restricted to a thickness of approximately 10 nm, they have limited the size of the functional integration of transmembrane protein. Other polymer cushions based on maleic anhydride copolymer thin films (from ethene, propene, and octadecene monomer units to poly(ethene-alt-maleic anhydride) (PEMA), poly(propene-alt-maleic anhydride) (PPMA), and poly(octadecene-alt-maleic anhydride) (POMA), respectively) have changed the lipid mobility in the membrane, depending on the hydrophilicity of the polymer cushion [328,329]. For example, the hydrolyzed thickness of POMA cushion is 4 nm, whereas it is 60 nm for the PEMA cushion. The fastest diffusion of lipid molecules on the PEMA cushion membrane is 1.24 µm^2^/s, whereas the diffusion on PPMA and POMA is 0.6 and 0.26 µm^2^/s, respectively. When the β-amyloid cleaving enzyme (BACE), which plays an active role in Alzheimer’s Disease, is incorporated to those polymer-cushioned lipid bilayers with varying physicochemical properties, the cushioning of supported lipid membrane leads to an increase in the incorporation and enzymatic activity of the reconstituted BACE with a direct correlation between lipid mobility and BACE activity [330]. Moreover, polyelectrolyte cushions composed of multilayers of PAH and PSS have been adsorbed on functionalized substrates, followed by the fusion of lipid vesicles made of dimyristoyl-l-α-phosphatidylglycerol (DMPG) and DMPC to complete polyelectrolyte-cushioned membranes [296].

PEG is the most commonly used polymer cushion and can create a reservoir as thick as 10 nm under the membranes [323], and thus such a platform is preferable for the incorporation of transmembrane proteins. For the PEG-cushioned membranes, the hydration level is reported as approximately 90%, and lateral mobility of lipid is 2.1 μm^2^/s, suggesting the formation of rugged bilayer membranes [331]. PEG is also attached to the silica substrate with silane terminal groups, and then it is completed with the lipid bilayer. The obtained membrane hosts the cytochrome C. Irrespective of tethering density of PEG, 80% of proteins in the membrane have retained the lateral mobility of protein and have been reported according to three classes—fast, slow, and immobile. Moreover, PEG-cushioned lipid bilayers derived from cell plasma membranes have been created to characterize the dynamics of single membrane proteins (e.g., glycosylphosphatidylinositol (GPI)-anchored protein, single-pass transmembrane glycoprotein CD98 heavy chain, and seven-pass transmembrane somatostatin receptor type 3 (SSTR3) protein with G-fluorescent protein (GFP) tag depending on the PEG length) (Figure 12A) [332].

In addition to GFP tagged proteins, dynamics of dye-labeled lipid, ATTO532-1, 2-dioleoyl-*sn*-glycero-3-phosphoethanolamine (DOPE) have been used as control. The fluorophore-labeled membrane molecules have appeared as bright spots in the fluorescence image (Figure 12B), and trajectories have been obtained from fluorescence videos by fluorescence single-molecule tracking (Figure 12C). From every trajectory, the diffusion coefficient has been estimated. The PEG with molecular weight from 1000 to 5000 g/mol has enhanced the mobile fraction of the membrane proteins. The diffusion coefficients of transmembrane proteins (CD98-GFP and SSTR3-GFP) have also been increased by a polymer cushion with increasing PEG length. However, the diffusion coefficient of the GPI-anchored protein has remained almost identical for different polymer lengths. Importantly, the diffusion coefficients of the three membrane proteins have become alike (2.5 μm^2^/s approximately). However, the cushioned membrane with the highest weight of 5000 g/mol has indicated that at the microscopic scale, the membranes have been fully suspended from the solid support by the polymer cushion.

### 4.3. Properties and Applications of Polymer-Lipid Composite Films

Model membrane platforms play a crucial role in unraveling the fundamental cellular processes that are involved in sensing and screening for pathogens or pharmaceutical targets. To date, a commercially available sensor based on the electro-optical transducers is SPR since it enables us to report small changes on the sensor in real-time upon binding of analytes/proteins to immobilized ligand [333]. However, these sensors suffer from nonspecific interactions of analytes with the sensor surface. Therefore, it is necessary to cover the sensor surface with membrane platforms, which not only show non-fouling characteristics but also accommodate different membrane proteins without denaturing them. One promising strategy to design biosensors is to use polymer-lipid composite films since they provide high mechanical stability and electrical resistance in the order of MΩ·cm^2^, which are the key properties of membranes. Additionally, the electrical properties of the polymer-lipid composite films change depending on the choice of polymers and substrates, leading to different measurement configurations. For example, biosensors based on the direct read-out of conductance signals from ion channels or transport proteins incorporated into polymer-lipid composite films formed on ITO semiconductor electrodes have been developed [26,27].

Additionally, position selective detection of enzymes/proteins on polymer-lipid composite films by fluorescence microscopy is very attractive. Such systems can be obtained by the combination of array integrated or micro-patterned device and spatial confinement of polymer-lipid composite film with membrane proteins. For example, micro-patterned of polymer-lipid composite films with different membrane proteins (e.g., ion channels) are incorporated into the arrays of a field-effect transistor with a sensor area of tens of μm^2^. Then, such a platform allows parallel screening of the protein activity from individual film patches. This platform can be also combined with microfluidic devices that provide controlled delivery of molecules/analytes to each part of arrays, resulting in high throughput screening [334,335]. Moreover, combining microfluidic systems with biomolecules-functionalized polymer-lipid composite film arrays can be also readily applied in clinical diagnostics to detect low abundant predefined proteins in blood/serum or urine for the early stage of diagnosis of a disease.

Polymer-lipid composite films can be also used as a cell culture platform since solid-supported model membrane platforms have been initially designed to study cell-cell recognition in the immune system [336]. For example, T-lymphocytes and neutrophils have been already sued for immunological studies on supported model membrane platforms [337,338]. It is also possible to tune the cell behavior by either adjusting lipid composition [339] or changing reconstituted membrane proteins, e.g., GPI-modified E-cadherin proteins [45] or peptides, e.g., RGD peptides [340] in the lipid-polymer composite films. Since the RGD peptides are responsible for controlling cell adhesion and growth, mouse fibroblast cell adhesion has been blocked on the supported lipid membranes; however, it has been promoted in the presence of RGD on the membrane surface [340].

## 5. Hybrid Polymer-Lipid Supported Membranes

Hybrid membranes composed of amphiphilic block copolymers and phospholipids represent a promising class of materials for the development of artificial membranes. Since lipid bilayers are relatively unstable compared to commercially available membranes, synthetic bilayers are favored due to their improved physicochemical properties, i.e., high mechanical and chemical stability, as well as flexibility [7,341,342,343], low water and gas permeability [344], and customizable properties, e.g., a larger range of membrane thickness [7] and end groups [345,346], while mimicking the architecture and applications of biological membranes, e.g., pore insertion [347,348,349]. Amphiphilic block copolymers have been shown to assemble into bilayer-like structures [7,350]. They are most commonly synthesized for biomimetic applications as either diblocks with a hydrophilic and a hydrophobic block or as triblock copolymers with two hydrophilic blocks and one hydrophobic middle block.

Having intermediate characteristics between pure copolymer and pure lipid bilayers [7], polymer-lipid hybrid membranes can be homogeneous, if the polymer and lipid components are completely miscible, otherwise heterogeneous, if they are immiscible, leading to form phase domains. Phase domain separation plays an important role in the process of biomolecule-membrane combination, making hybrid membranes of particular interest for mimicking biological membranes, e.g., for studying the mechanisms of membrane related processes [6] or development of biomimetic platforms for biosensing and drug delivery applications.

### 5.1. Assembly of Hybrid Membranes Based on Polymers and Lipids

Up to date, copolymer-lipid-based hybrid membranes have been assembled in a planar fashion using different deposition strategies. They can be deposited onto solid support in the form of two-dimensional supramolecular assemblies, i.e., mono-, bi-, or multi-layers with a membrane-like architecture. The typical substrates employed for their deposition include SiO_2_, Au, glass, and mica, to prepare membranes of different hydrophobicity/hydrophilicity properties [297,298,299,351]. The most common methods used for the formation of solid-supported membranes are (i) vesicle fusion (ii) LB techniques (Figure 8) [352], and (iii) solvent-assisted bilayer formation [353,354,355]. All these methods are based on physical interactions (hydrophobic/hydrophilic) between the lipid, the polymer, and the substrate, which drive their self-assembly into membranes [356]. Vesicle fusion results in the deposition of vesicles onto solid support that fuses into a bilayer planar membrane [351,357]. Due to the simultaneous influence of various parameters, such as pH, ionic strength, the chemical composition of the polymer, or size and distribution of the vesicles, it is challenging to control the properties of the films obtained using this method [297,352]. The main factors to overcome in order to achieve the opening and the fusion of vesicles containing polymers are their higher mechanical stability compared to lipid vesicles, the unfavorable thermodynamic conditions, and the loss of conformational freedom derived by the packing of the membrane. Polymer-lipid hybrid vesicles may show a better tendency to rupture than polymer vesicles due to the properties of the lipids being less rigid. Polymer-lipid supported membranes have been obtained by the fusion of hybrid vesicles triggered by osmotic shock in a study, making them in the controlled and reproducible manner [356]. Regardless of the content of polymer and lipid in the hybrid mixture, it has been possible to produce a membrane with both enhanced mechanical stability and mobility compared to one-component membranes [6], which could be explained with a synergic activity of polymer and lipid in the two-component hybrid membrane.

In the LB method, the two-component lipid-polymer monolayers typically exhibit intermediate characteristics between the single monolayers of the component polymers and lipids in terms of breaking point and phase transition pressure values [6]. However, the LB method is less favorable for hybrid membrane formation due to the requirements of costly equipment and ultra-experimental conditions. Additionally, the solvent-assisted lipid bilayer (SALB) method consists of the deposition of a lipid dissolved in an organic solvent on a solid support, followed by an exchange of the solvent with an aqueous buffer (Figure 13) [354]. This method can be applied also to the polymer and polymer-lipid mixture with proper adaption. The SALB method overcomes the limitation of vesicle rupture to form bilayers onto a limited number of substrates. As reported in a comparison study between vesicle rupture and SALB methods, only with the latter, it has been possible to successfully obtain bilayers onto silicon dioxide, gold, and alkanethiol-functionalized gold (Figure 13). However, the SALB approach has the limitation of a required solubility of the chemical compound in an organic solvent. Recently, it has been attempted for the first time to prepare a lipid-polymer membrane on a crystalline layer of proteins [358]. Blending the polybutadiene (PBD)-PEO and 1,2-ioleoyl-*sn*-glycero-3-phosphocholine (DOPC) in specific chemical composition (molar ratio 3:7) allows creating a hybrid membrane with improved viscoelastic properties and a retained permeability. Interestingly, the hybrid membrane has behaved as a barrier for the protein layer against the attack of an enzyme [359,360].

Polymer-lipid hybrid membranes are characterized in a similar way to PE-based hybrid films. Additionally, Langmuir isotherm and BAM are employed to monitor the monolayer formation at the air–water interface. By means of QCM-D, it is possible to characterize the hybrid membrane in terms of viscoelastic properties and membrane coverage using the QCM-D sensor as the solid support. Sessile droplet method and ellipsometry can be further employed to determine the hydrophobicity/hydrophilicity of the membrane and its thickness, respectively.

### 5.2. Properties and Applications of Polymer-Lipid Membranes

Profiting from the properties of phospholipids and copolymers, one can design and develop hybrid membrane platforms with advanced features [47,180,361]. Polymers possess good mechanical resistance and, if appropriately selected, their lateral mobility within the membrane is significant, even enabling the recombination with membrane proteins [247,362]. On the other hand, the presence of the lipids can enhance the mobility of the membrane and support the specific interaction with membrane proteins [351]. The physical state of the hybrid components, either gel or liquid phase, determines the protein distribution within the membrane, as already reported [6]. In fact, it has been observed that when a variety of block copolymers and lipids have been combined, they have revealed different properties in hybrid membranes as fluidity or capability of promoting the insertion of biomolecules, such as proteins. For example, the interactions between two lipids, dipalmitoylphosphatidylcholine and 1,2-dioleoyl-*sn*-glycero-3-phosphocholine (DPPC and DOPC), and poly(dimethylsiloxane)-block-poly(2-methyl-2-oxazoline)-block-poly(dimethylsiloxane) (PMOXA-*b*-PDMS-*b*-PMOXA) copolymers monolayers have been characterized by monitoring the energetic effects related to the mixing/demixing processes. The lipids have been selected because of their different way of packing properties within the membrane, namely, their self-assembly into gel or fluid phase monolayers. The diversity in shape and dimension of domains in the observed hybrid films have depended mainly on the packing behavior of the molecules, related to the size of their head groups, and/or on the length mismatch between the polymer and the lipid, serving to control the fluidity of the membrane [6].

Besides, diblock copolymers, such as PDMS-*b*-PMOXA, have been combined with various phospholipids found in cell membranes, i.e., DPPC, DOPC, and 1,2-dipalmitoyl-*sn*-glycero-3-phosphoethanolamine (DPPE), in diverse molar ratio values, in order to investigate the factors inducing the domain separation [6]. The surface pressure applied to the hybrid monolayer has played a decisive role in controlling the domain formation. When diblock copolymer and DPPC have been combined, the lipid domains have formed above the phase transition point, namely, at the surface pressure where it assumes a gel phase. Moreover, the length of the polymer hydrophobic chains has affected the lipid domains in terms of stabilization of the shape and control over the size. As a first attempt to study how block copolymers influence the surface rheology of a hybrid film, DPPC has been combined with PEO-poly(butyleneoxide) (PBO) [180]. Monolayers of diblock copolymers at air–water interface typically adopt a brush conformation, where the hydrophilic block hangs in the water, and the hydrophobic one stays anchored to the surface. The isotherm of the PEO-PBO monolayer has shown a transition from expanded to the condensed phase when compressed, and in the case of the mixed monolayer with DPPC, the two species have remained phase separated. In this case, a negligible interaction between the polymer and the lipid has been observed with no enhancement of the monolayer mechanical resistance. This makes PEO-PBO an excellent candidate for polymer addition into lipid membrane with almost no change in the lipid architecture, a crucial requirement in the field of biomedical applications.

Polymer film undergoes structure reorganization after the addition of a small amount of lipid, due to their repulsive interactions. The driving conditions for obtaining heterogeneous membranes involve the structure and the phase transition temperature of polymer and lipid, together with the molar ratio of their mixture [47,363,364]. The temperature drives the kinetics of domain clustering: small-sized domains tend to aggregate into larger domains for when the temperature is higher than the melting point (T_m_) of the lipid, in favor of a more thermodynamically stable system [365]. In addition, polymer-lipid phase separation does not completely occur, but domains partially overlap each other; “domain” refers, in fact, to either a lipid- or polymer-enriched area [6]. The molar ratio is a key factor to determine the phase domain distribution within the membrane: for a concentration of one species in the hybrid—either polymer or lipid—below 50%, it has aggregated into the upper or lower leaflet, whereas, above 50%, the phase separation has involved both leaflets of the membrane. An in-depth computational study of phase domain separation in hybrid membranes consisting of A_3_B_x_ (x from 11 to 25) diblock copolymers and model phospholipids in gel and fluid phase [365] has shown how the lipid unsaturation, the mismatch between lipids and polymers, and the consequent energy incompatibility between their hydrophobic parts could influence the formation of domains, as well as how the length of the polymeric hydrophobic chain influences the fluidity of the lipid part. The polymer-lipid interactions and the phase domain separation have been investigated in different monolayers composed of polyisobutylene (PIB), PEO, and DMPC [363], revealing that some polymer or lipid molecules have been forced during the compression to diffuse into a different domain, resulting in the formation of lipid- and polymer-rich domains. The adsorption of poly(glycerol monomethacrylate)-*b*-poly(propylene oxide)-*b*-poly(glycerol monomethacrylate) (PGMA-*b*-PPO-*b*-PGMA) triblock copolymer onto presynthesized DPPC and DMPC model monolayer films [366] has been favored by the capability of the lipid monolayer to incorporate the polymer. The differences in the polymer have been related to the LE and LC phase transition of the lipid, revealing the importance of this factor for an efficient phase domain separation, analogous to lipid rafts found in cell membranes. Lipid rafts are involved in many biological processes, such as signal transduction or protein lateral organization [367]. Controlling the phase domain separation allows to create membranes of desired properties and specific functions and control the amount and position of the biomolecules being incorporated into the membrane [6]. The capability of hybrid polymer-lipid membranes to undergo a rearrangement of their architectures in response to an external stimulus, e.g., upon contact with a biomolecule or a cell membrane, makes them applicable as a robust model for understanding membrane processes or investigating the biomolecule-membrane interactions. For example, polyhydroxylalkanoates (PHA) have been combined with DOPC for preparing monolayers at air–water interface as a model to analyze the molecular interactions between synthetic and biological membranes in terms of mixing/demixing properties [46]. Also, the incorporation of polymers into lipid-based systems can be a promising approach to improve the oral delivery of poorly water-soluble compounds, while the technique to incorporate the polymer plays an important role in affecting the bio-performance [368]. The insertion of the biomolecule into the polymer-lipid membrane can be driven by specificity (e.g., receptor/ligand recognition [369]) or occur through a spontaneous insertion. In the former case, the great diversity of functional groups possible to incorporate into polymers allows the covalent bonding of a chosen biomolecule [352]. In case of the insertion [370], a model system based on PMOXA-*b*-PDMS-*b*-PMOXA triblock copolymer-DPPC has indicated that the positioning of biomolecules (membrane proteins) in a complex thin-film environment could be regulated by the phase behavior of the film components: the partial separation between an amphiphilic polymer and a lipid has driven the protein to the fluid phase in a similar manner as in the case of cellular bilayers. The potassium channel from *Mesorhizobium loti* (MloK1) has been inserted into hybrid supported membranes obtained via blending PDMS-*b*-PMOXA with different lipids (1-palmitoyl-2-oleoyl-*sn*-glycero-3-phosphocholine (POPC), DPPC or 1-palmitoyl-2-oleoyl-*sn*-glycero-3-phosphoethanolamine (POPE)). The presence of the fluid POPC, combined with the polymer, forces the protein to insert spontaneously into the lipid phase. As for the homogeneous mixture, if PDMS-PMOXA and POPE, the protein has distributed into the whole film. Finally, when the PDMS-PMOXA-DPPC monolayer has undergone protein insertion, it has exhibited instead a preference toward the polymeric domain (Figure 14) [6]. It has also been observed that a high degree of fluidity is a key requisite for facilitating protein insertion into the membrane.

Phosphatidylserine and phosphatidylcholine (PC) membranes have been combined with PAH and PSS multilayers to allow the incorporation of colloidal nanoparticles with specific biological properties based on their content of viruses (Influenza A/PR8) or virus-like particles (Rubella) [372]. These are used as models for studying the virus integration mechanisms into supported membranes. Due to the strong interaction between the polymers and the lipids, it has been possible to controllably perform a 2D assembly and create a biocompatible membrane. The lateral diffusion of lipids during the integration has been used for the virus to study the virus-membrane interactions. This novel system has been able to produce a platform that can be applied for modification and design of artificial peptide epitopes on the virus species and can be extended onto a variety of other virus-based systems for mimicking virus infection pathways.

Due to the capability to mimic natural membranes and to be studied as models for understanding cellular membrane functions, hybrid membranes composed of phospholipids and copolymers can be utilized for a broad range of surface-specific applications in the field of, e.g., biotechnology and nanomedicine: biosensing, antimicrobial coatings, signal transduction, investigation of complex processes, substrate-mediated drug delivery, or controlled drug permeation into membranes [6,373,374]. Hybrid membranes are also used for the reconstitution of biomolecules [6,374,375,376]. For example, PBD-*b*-PEO/DPPC hybrid films have been employed to accommodate the anticancer drug Paclitaxel for drug delivery applications [369]. The presence of several micro- and nano-domains in the hybrid has served to load the drug in the domain boundaries with a resulting improved loading performance compared to the polymer or lipid membrane. A system composed of PEO-*b*-PBO and DPPC monolayers has been developed for its potential applications in the fields of controlled drug delivery or medical imaging, based on the polymer biocompatibility and surface activity [16]. PEO-*b*-PBO has been found to be a better candidate than poloxamer in terms of mixability with the cell membrane, mainly due to its diblock structure, lower polydispersivity, and lower surface tension [16]. Hybrid materials composed of PEO-*b*-PBO and ganglioside-functionalized DPPC have been employed for molecular recognition of cholera toxin B due to its specific interaction with the ganglioside [369]. A synthetic cationic lipid dioctadecyldimethylammonium bromide (DODAB) has been trapped within different polymer networks for the development of antimicrobial coatings, with the lipid component being the active agent [374]. The porosity of the structure observed, when DODAB is mixed with PS, has depended on the phase separation, while it has resulted homogeneous in the DODAB-PMMA. The presence of the polymer matrix has improved the functionality of this platform in terms of the interaction of the bactericidal DODAB with the virus: first high effectiveness has been proved with a lower amount of DODAB; second, the compatibility with the polymer matrix has prevented DODAB from leaking from the film. A solid-supported multilayered membrane composed of diblock copolymer poly(butadiene-*b*-ethylene oxide) (PBD-*b*-PEO) and DPPC [377] has improved the synergic permeability of a drug in comparison to single-component membranes. In the field of bio-sensing, hybrid membranes have been employed as taste sensors for the detection of bitterness [28,378,379], sweetness [29], or saltiness [30] in substances. The taste sensor utilizes lipid/polymer membranes, and the interaction with taste substances causes a change in the membrane potential, which can be monitored. These chemical sensors serve to mimic the human sense, and generally, they consist of polymers, lipids, and plasticizers, which have, respectively, the function to stabilize the membrane and to control its hydrophobicity and its electrical properties. The membrane itself has been composed of polyvinylchloride (PVC) as polymer and tetradodecylammonium bromide (TDAB) as lipid, with a concentration of lipid tuned to achieve selective detection against a specific taste. Interestingly, it has been possible to change the surface structure of the hybrid membrane through a process called “preconditioning”, with the purpose of controlling the amount of binding of the wanted molecule [28,378]. For example, inducing a membrane change from hydrophobic to hydrophilic, it has been possible to bind a different amount of the *iso*-alpha acid contained in beer and responsible for its bitter taste [28]. Thus, the amount of adsorption of a substance onto the hybrid membrane depends on the surface charge density (related to the concentration of a charged lipid present on the membrane), as well as on the membrane saturation, the temperature, and the membrane porosity [380]. In another study, a hybrid membrane composed of TDAB and di-*n*-octylphenylphosphonate (DOPP) lipids polymerized with derivates of tryhydroxybenzoic acid and with high sensitivity to sugar has been developed and modified with phenolic compounds of the different substituted position of hydroxylic groups, influencing its responsiveness towards the taste molecule [29]. In addition to the conventional system, which does not present ionophores, their addition—selective to saltiness-enhancing molecules (e.g., NaCl)—has been found to enhance the response of the sensor [30].

## 6. Conclusions

Hybrid biomimetic polymer-based films are created by assembling the biological elements (e.g., lipids, proteins, peptides, and DNA) and the polymers (e.g., polyelectrolytes, homopolymer, and amphiphilic block copolymers) on a solid support. Such hybrid films exhibit many fascinating properties depending on the polymer and biological elements. In general, polymer-based films with the polyelectrolytes or polymer brushes form the robust, stable, and tunable structural film matrices, whereas the biological elements are either attached or embedded into these matrices to induce biological functions. In polymer-lipid based films, the polymers are either used as a tether or cushion between solid support and lipid bilayer membrane or are blended with lipids to create hybrid lipid-polymer membranes. For these films, the biomolecules are either attached or incorporated into the lipid or hybrid membranes, respectively. Despite continuing challenges of incorporating the biological elements into the polymer-based films and observing their functional properties with different characterization methods, the ability to harness these films improves day by day. Currently, all existing polymer-based films with different biomolecules not only allow us to understand biofunctionality of synthetic matrices but also to develop active smart surfaces to use in many different biotechnological or translational applications from biosensing to high throughput screening, cell-culture platforms to anti-biofouling coatings, biomimetic lubricants to catalytic matrices. It is likely that the future generations of hybrid biomimetic polymer-based films will involve more complexity in terms of their structural compositions (e.g., polymer and biomolecules) and functional outcomes that can be measured by characterization methods by tracking multiple electrical, optical, and acoustic signals. Future developments in polymer-based films with practically G-protein coupled receptors (GPCRs) that are associated with many diseases could proportionate many unique and beneficial applications in the pharmaceutical industry and could significantly advance the treatment of GPCRs-associated diseases.

## List of Abbreviations

**AAO** porous anodic aluminum oxide

**Ab2** secondary antibodies

**AChE** acetylcholinesterase

**AFM** atomic force microscopy

**AFP** alpha-1-fetoprotein

**anti-IgG** IgG antibodies

**APCs** antigen presenting cells

**ATP** adenosine triphosphate

**ATRP** atom transfer radical polymerization

**BACE** β-amyloid cleaving enzyme

**BAM** Brewster angle microscopy

**bFGF** basic fibroblastic factor

**B-PC-PEI** biotinylated photocleavable polyethylenimine

**BSA** bovine serum albumin

**ChO** choline oxidase

**CLSM** confocal laser scanning microscopy

**CSA** chondroitin sulfate

**CTMA** cetyltrimethylammonium chloride

**CV** variation coefficient

**DEPN***N*-tert-butyl-*N*-1-diethylphosphono-2,2-dimenthylpropylnitroxyl

**DMA** dimethylmaleic anhydride

**DMAEMA** 2-(dimethylamino) ethyl methacrylate

**DMPC** dimyristoylphosphatidylcholine

**DMPG** dimyristoyl-l-α-phosphatidylglycerol

**DNA** deoxyribonucleic acid

**DODA** Dioctadecylamine

**DODA-E85** Dioctadecylamine [poly(ethyl-oxazoline]

**DOPC** 1,2-dioleoyl-*sn*-glycero-3-phosphocholine

**DOPE** 1,2-dioleoyl-*sn*-glycero-3-phosphoethanolamine

**DOPP** di-*n*-octylphenylphosphonate

**DPPC** dipalmitoylphosphatidylcholine

**DPPE** 1,2-dipalmitoyl-*sn*-glycero-3-phosphoethanolamine

**E. coli** Escherichia coli

**EDC** 1-ethyl-3-(3-dimethylaminopropyl)carbodiimide

**EIS** electro-impedance spectroscopy

**ELISA** enzyme-linked immunosorbent assay

**EPR** electron paramagnetic resonance

**ETM** electron transfer material

**FESEM** field-emission scanning electron microscopy

**FITC** fluorescein isothiocyanate label

**FRAP** fluorescence recovery after photobleaching

**FRAP** fluorescence recovery after photo-bleaching

**FRET** fluorescence resonance energy transfer

**FTIR** Fourier transform infrared spectroscopy

**GCE** glassy carbon electrode

**GFP** green fluorescent protein

**GOx** glucose oxidase

**GPCRs** G-protein coupled receptors

**GPI** Glycophosphatidylinositol

**GUMA** deoxy-2-*N*-(2′-methacryloyloxyethyl) aminocarbamyl d-glucose

**HA** hyaluronic acid

**Hb** hemoglobin

**HEPG2** human liver cancer cell line Hep G2

**HEK93** human embryonic kidney 293 cells

**HIV** human immunodeficiency virus

**HRP** horseradish peroxidase

**HSA** human serum albumin

**IgG** goat immunoglobulin G

**IL-2** interleukin 2

**IPM** interphotoreceptor matrix

**ITO** tin oxide

**LB** Langmuir-Blodgett

**LbL** layer-by-layer

**LC** liquid condensed

**LE** liquid expanded

**LOD** lower detection limit

**LS** Langmuir-Schaeffer

**MloK1** Mesorhizobium loti

**MMA** methyl methacrylate

**MN** microneedle arrays

**MS** mass spectroscopy

**MSA** methanesulfonic acid

**MUA** 11-mercaptoundecanoic acid

**MWCNT** multi-walled carbon nanotubes

**NAs** nerve agents

**NIPAAM***N*-isopropylacrylamide

**NMMO***N*-methylmorpholine oxide

**NMP** nitroxide-mediated polymerization

**NMR** nuclear magnetic resonance

**NPs** nanoparticles

**OPP** organophosphate pesticide

**OSM** oligo(sulfamethazine)

**PAA** poly(acrylic acid)

**PAAH** polyallylamine hydrochloride

**PAEU** poly(amino urethane)

**PAH** poly(allylamine hydrochloride)

**PAMAM** polyamidoamine

**PANI** polyaniline 

**PBA** poly(n-butyl acrylate)

**PBA** poly(n-butyl acrylate)

**PBD** poly(butadiene)

**PBMA** poly(n-butyl methacrylate)

**PBO** poly(butyleneoxide)

**PC** phosphatidylcholine

**PCBAA** poly-carboxybetaine acrylamide

**PDA***o*-phenylenediamine

**pDA** polydopamine

**PDADMAC** poly(diallyldimethylammonium chloride)

**PDDA** poly(diallyldimethylammonium chloride)

**PDMAEMA** poly(*N*,*N*-dimethylaminoethylmethacrylate) ()

**PDMMA** poly[2-(dimethylamino)ethyl methacrylate-block-methyl methacrylate]

**PDMS** poly(2-methyl-2-oxazoline)

**PDMS** polydimethylsiloxane

**PDMSB** poly(1,1-dimethysilacyclobutane)

**PE** polyelectrolytes

**PED** polymeric enzyme detection

**PEDOT-S** poly(4-(2,3-dihydrothieno[3,4-b][1,4]dioxin-2-yl-methoxy)-1-butanesulfonic acid

**PEG** poly(ethylene glycol)

**PEI** poly(ethylene imine)

**PEMA** poly(ethene-alt-maleic anhydride)

**PEO** polyethylene glycol

**PEOX** poly(ethyl oxazoline)

**PET** polyethylenterephthalate

**PFN** poly{9,9-di[3-(1-ethyl-1,1-dimethyl ammonio)propyl]-2,7-fluorenyl-alt-1,4-phenylene dibromide}

**PFPBr** poly[(9,9-bis(6′-*N,N,N*-trimethylammonium)hexyl) fluorene-alt-1,4-phenylene] bromide)

**PGMA** poly(glycerol monomethacrylate)

**PHA** polyhydroxylalkanoates

**PHEMA** poly-phenoxyethylmethacrylate

**PHPMA** poly-hydroxypropylmethacrylate

**PIB** polyisobutylene

**PIMP** photoiniferter-mediated polymerization

**PL** photoluminescence

**PLA** polylactide

**PLGA** poly(lactic-co-glycolic acid)

**PLL** poly(L-lysine)

**PMMA** poly(methylmethacrylate)

**PMOXA** poly(dimethylsiloxane)

**PMPC** poly(2-methacryloyloxyethyl phosphorylcholine)

**PNA** peptide nucleic acid

**PNMP** poly(o-nitrobenzyl-methacrylate-co-methyl-methacrylate-co-poly(ethylene-glycol)-methacrylate)

**pNIPAAM** poly(*N*-isopropylacrylamide)

**POEGMA** poly(oligo (ethylene glycol) methacrylate)

**POMA** poly(octadecene-alt-maleic anhydride)

**PPMA** poly(propene-alt-maleic anhydride)

**PPO** poly(propylene oxide)

**PS** poly(styrene)

**PSBMA** poly(sulfobetaine methacrylate)

**PSS** poly (4-styrenesulfonicacid) sodium salt

**PtBA** poly(tert-butyl acrylate)

**PTMPHFs** trimethylphosphonium-substituted polyfluorenes

**PVC** polyvinylchloride

**PVS** poly(vinylsulfonic acid, sodium salt)

**QCM-D** quartz crystal microbalance with dissipation monitoring

**RAFT** reversible addition-fragmentation chain transfer

**RGD** arginylglycylaspartic acid

**S. epidermidis** Staphylococcus epidermidis

**SALB** solvent-assisted lipid bilayer

**SAMs** self-assembled monolayers

**SC** stratum corneum

**SEM** electron microscopy

**SOD** superoxide dismutase

**SPMA** spiropyran-containing methacrylate

**SPR** surface plasmon resonance

**ssDNA** single stranded DNA

**SSTR3** somatostatin receptor type 3

**TDAB** tetradodecylammonium bromide

**TEMPO** 2,2,6,6-tetramethyl-piperidinyloxy

**TIPNO** 2,2,5-trimethyl-4-phenyl-3-azahexane-3-oxyl

**T_m_** melting point

**TMB** 3,3’,5,5’-tetramethylbenzidine

**TNF-α** tumor necrosis factor α

**UV** ultraviolet

**VBTAC** vinylbenzyl-trimethylammonium chloride

**XPS** X-Ray photoelectron spectroscopy

## Figures and Tables

**Figure 1 polymers-12-01003-f001:**
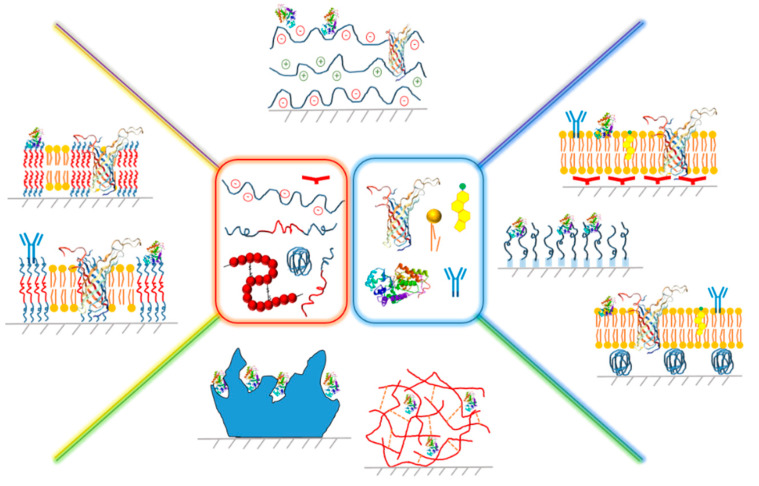
Schematic presentation of biomimetic hybrid films based on the combination of biomolecules (enzymes, proteins, nucleic acids, natural polymers, and lipids) and different polymer species: polyelectrolytes (**top**), polymeric structures (brushes, tethers, and cushions) (**right**), synthetic polymers (**bottom**), and amphiphilic block copolymers (**left**).

**Figure 2 polymers-12-01003-f002:**
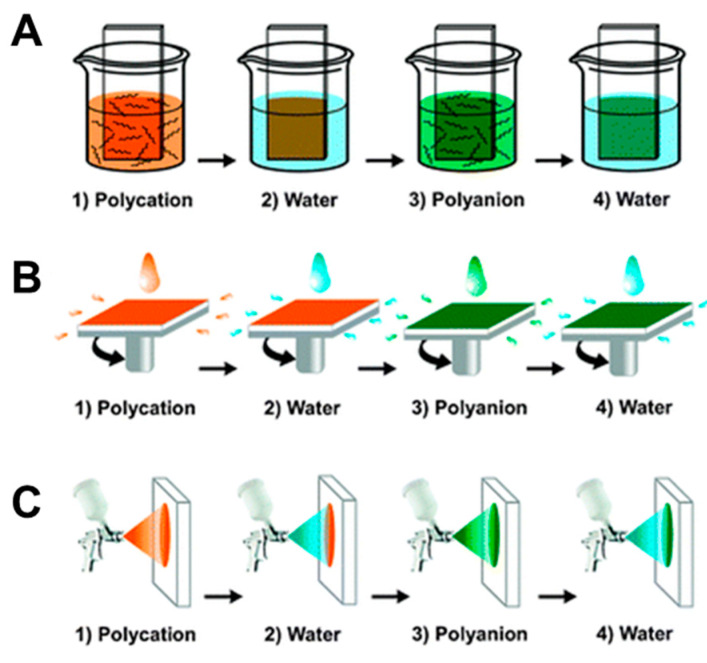
Schematic representation of the processes used to fabricate polyelectrolyte multilayer films by layer-by-layer (LbL) assembly. (**A**) Dip coating: glass slides and beakers are used in this method. Steps (1) and (3) represent the exposure of a polyanion and polycation, respectively, and steps (2) and (4) are rinsing steps. The four steps are in the basic build-up sequence and contribute to only one-bilayer film architecture. If the 10-bilayer film is needed, the four procedures would be repeated 10 times. Construction of more complex film requires additional beakers and an extended deposition sequence. (**B**) Spin-assisted LbL assembly: High spinning speed would be operated after the droplet of the material is applied onto the center of the substrate surface. The rinsing step would take place between steps (1) and (3), and all the four steps are in a circulation system. (**C**) Spray-assisted LbL assembly: Instead of bringing the substrates’ surface into contact with the liquid of the adsorbing species, the liquid is sprayed against the receiving surface of the substrates. Eight multilayer films would be formed by repeating steps (1) to 94) in a cyclical fashion. Reproduced from Li et al. (2012) [70] with permission (Copyright © The Royal Society of Chemistry 2012).

**Figure 3 polymers-12-01003-f003:**
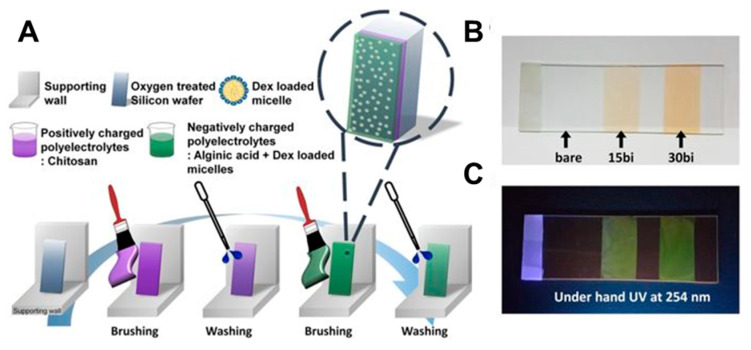
(**A**) Schematic illustration of the brushing layer-by-layer (LbL) self-assembly process on a flat substrate. Photographs of the (PAH-FITC/PAA)_n_ (n  =  number of bilayers) multilayer films prepared by brushing LbL on a glass slide under (**B**) visible light and (**C**) ultraviolet (UV) light (λ  =  254) (Figure 3A is representing brushing LbL process, and it was drawn by K.P.). Reproduced from Kyungtae et al. (2018) [71] with permission (Copyright © Springer Nature 2018). PAH, poly(allylamine hydrochloride); PAA, poly(acrylic acid).

**Figure 4 polymers-12-01003-f004:**
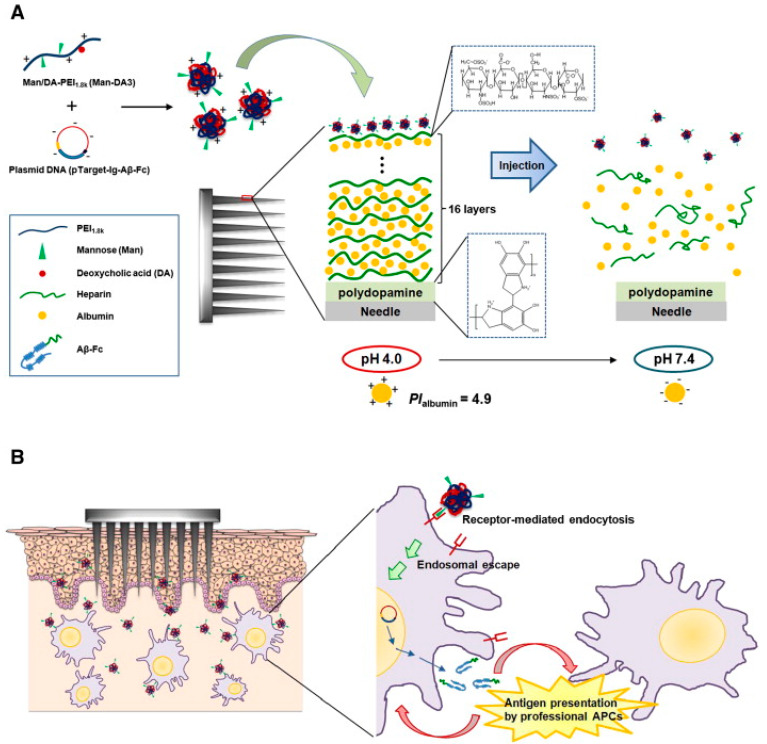
(**A**) Schematic illustration of the release of functional polyplexes from microneedle arrays (MNs) coated with pH-responsive polyelectrolyte multilayer assembly upon application to the skin. (**B**) Targeted delivery of polyplexes with surface mannose moieties to intradermal resident antigen presenting cells (APCs) after release from the MNs. Reproduced from Kim et al. (2014) [32] with permission (Copyright © 2014 Elsevier Inc. All rights reserved.).

**Figure 5 polymers-12-01003-f005:**
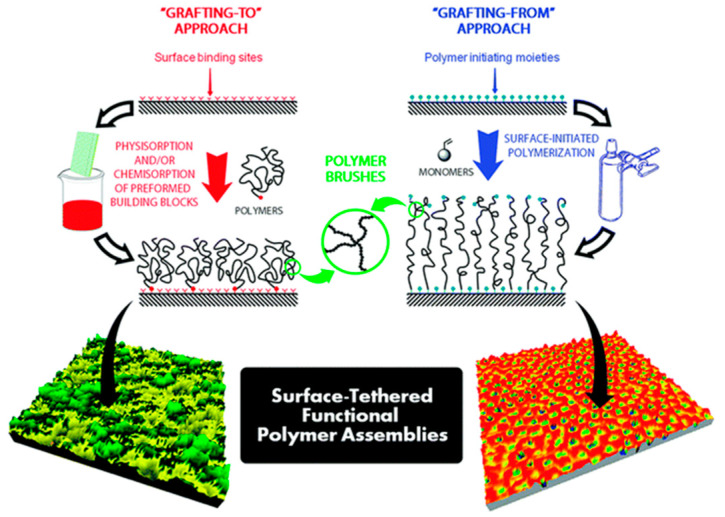
Schematic illustration describing main chemical strategies (grafting to and grafting from approaches) used to tether functional polymer brushes on different substrates. Reproduced from Giussi et al. (2019) [183] with permission (Copyright © The Royal Society of Chemistry 2019).

**Figure 6 polymers-12-01003-f006:**
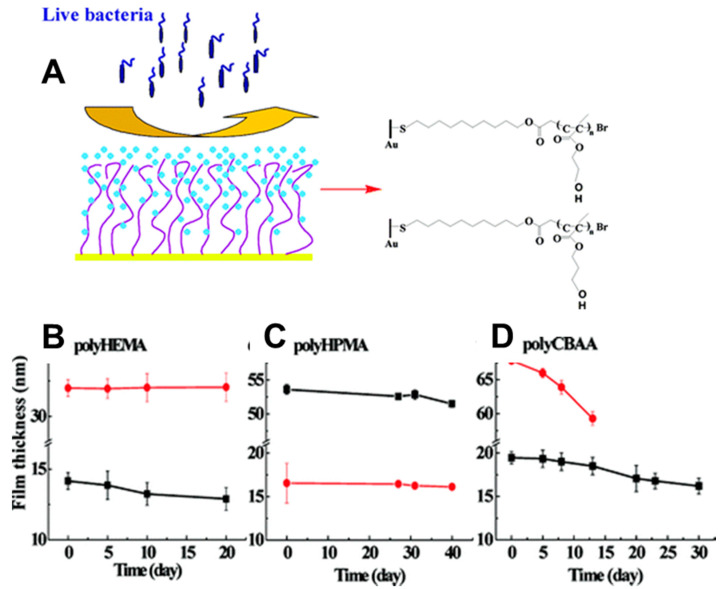
(**A**) Non-fouling biomaterial from gold substrate-initiated atom transfer radical polymerization (ATRP). Change of polymer film thickness of polyHEMA (**B**), polyHPMA (**C**), and polyCBAA (**D**) on surface plasmon resonance (SPR) chips as a function of incubation time in PBS solution (pH 7.4, 0.15 M, 138 mM NaCl, 2.7 mM KCl) for stability test [195]. Reproduced from Zhao et al. (2011) with permission (Copyright © American Chemical Society 2011). HEMA, phenoxyethylmethacrylate; HPMA, hydroxypropylmethacrylate; CBAA, carboxybetaine acrylamide.

**Figure 7 polymers-12-01003-f007:**
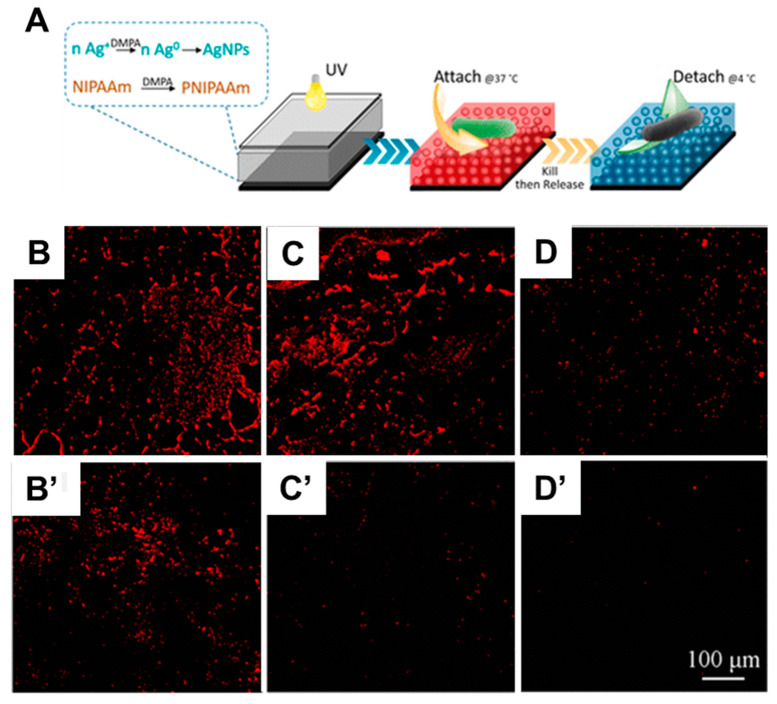
(**A**) Strategy for the synthesis of smart polymer surfaces by photopolymerization to provide for bacterial attachment and detachment in response to the change in environmental temperature, bottom: confocal microscope fluorescence micrographs of *Escherichia coli* on (**B**) glass, (**C**) pNIPAAM-glass, (**D**) Silver nanoparticles (AgNPs)/PNIPAAm-glass. (**B′**–**D**′) Confocal microscope fluorescence micrographs of *E. coli* after immersion in normal saline and shaking at 4 °C for 0.5 h (speed, 100 rpm) [224]. Adapted from Yang et al. (2016) with permission (Copyright © American Chemical Society 2016).

**Figure 8 polymers-12-01003-f008:**
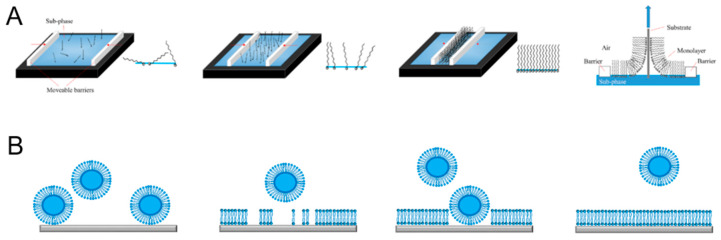
(**A**) Schematic showing the steps involved in the formation of Langmuir–Blodgett films. Each image shows the trough set-up and a side-on view of the interface. Amphiphile is spread onto the sub-phase on a Langmuir trough resulting in a 2D ‘gaseous’ arrangement of amphiphiles (i.e., no interactions between molecules). Barriers are compressed to reduce the surface area of the interface, and molecules begin to interact, forming a 2D ‘liquid expanded’ phase. On further compression, the amphiphiles are self-assembled into a monolayer, forming a 2D ‘liquid compressed’ phase. When a monolayer has formed, it can be transferred onto solid support via vertical deposition. Red arrows indicate barrier movement direction. (**B**) Schematic representation of lipid vesicle fusion; deposition/adhesion, vesicles deposit onto a substrate over time; rupture, vesicles rupture, and form patches; the coalescence of lipid patches; adjacent lipid patches merge to form larger patches while some vesicles are trapped; completion, trapped vesicles, and vesicles from solution fix the homogeneous single lipid bilayer [248]. Adapted from Wales et al. (2016) with permission (Copyright © Springer Nature 2016).

**Figure 9 polymers-12-01003-f009:**
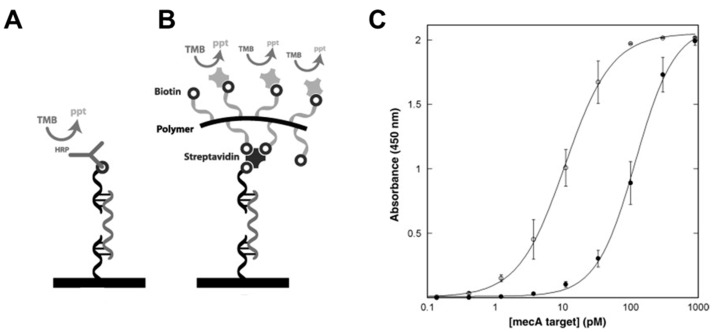
Schematic representation of standard detection and polymeric enzyme detection (PED)-enhanced detection of DNA hybridization. (**A**) Standard detection. The surface hybridized biotin-labeled oligonucleotide is detected by anti-biotin antibody/HRP conjugate reacted with the substrate—3,3’,5,5’-tetramethylbenzidine (TMB)—to produce a detectable signal. (**B**) PED detection. (**C**) The lower limit of detection of DNA hybridization using microtiter plate format. Various concentrations of a biotin-labeled, single-stranded target sequence from the mecA gene are hybridized to the mecA-specific capture probe immobilized onto microtiter wells. The signal is developed using standard detection (closed circles) or with the described PED polymer (open circles). Three experiments have been performed, each in triplicate (n = 9), and the mean absorbance reading ± 1 SD (error bars) is plotted [271]. Reproduced from Klonoski et al. (2010) with permission (Copyright © 2009 Elsevier Inc. All rights reserved.).

**Figure 10 polymers-12-01003-f010:**
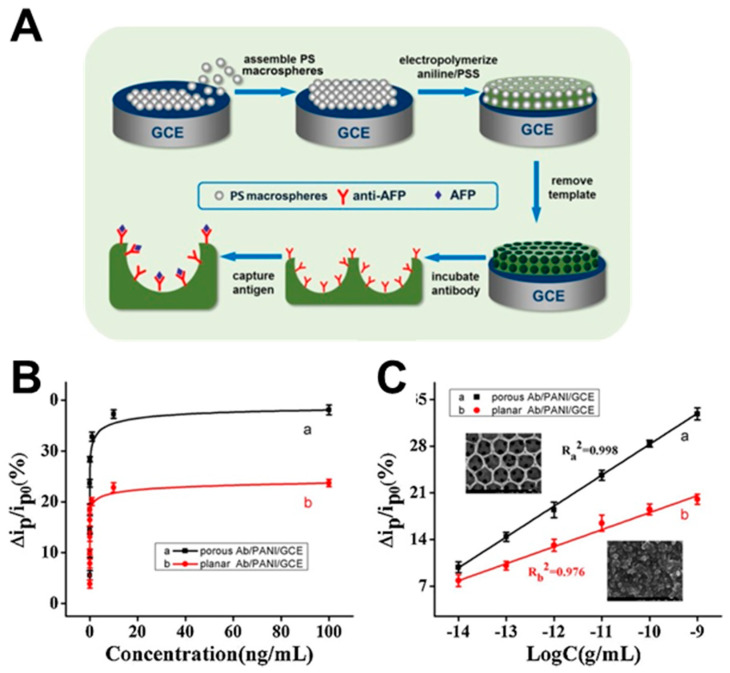
(**A**) Fabrication process of the alpha-1-fetoprotein (AFP) immunosensor based on macroporous polyaniline (PANI). (**B**) The dynamic response range of (a) porous PANI/ glassy carbon electrode (GCE) and (b) planar PANI/GCE upon a function of AFP concentrations. (**C**) The corresponding calibration curves of (a) porous PANI/GCE and (b) planar PANI/GCE. Error bars indicate the standard deviation of three measurements [273]. Reproduced from Liu et al. (2018) with permission (Copyright © 2017 Elsevier B.V. All rights reserved.).

**Figure 11 polymers-12-01003-f011:**
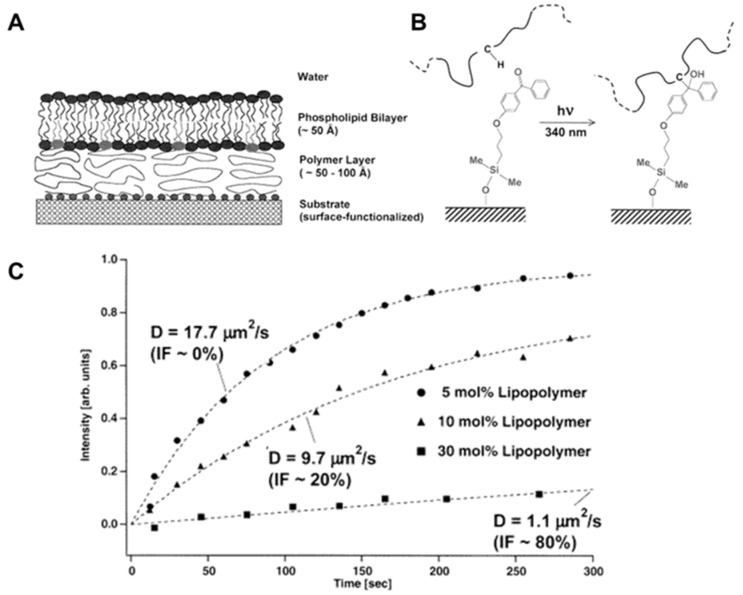
(**A**) Schematic illustration of the polymer-tethered phospholipid bilayer. (**B**) Schematic illustration of the photochemical attachment of polymer chains to solid substrates functionalized with the benzophenone silane photocoupling agent. (**C**) Fluorescence recovery curves of a polymer-tethered bilayer at different lipopolymer molar ratios of 5, 10, and 30 mol%, where only the outer leaflet of the bilayer is labeled. All data are recorded at T = 40 °C [314]. Reproduced from Naumann et al. (2002) with permission (Copyright © American Chemical Society 2002).

**Figure 12 polymers-12-01003-f012:**
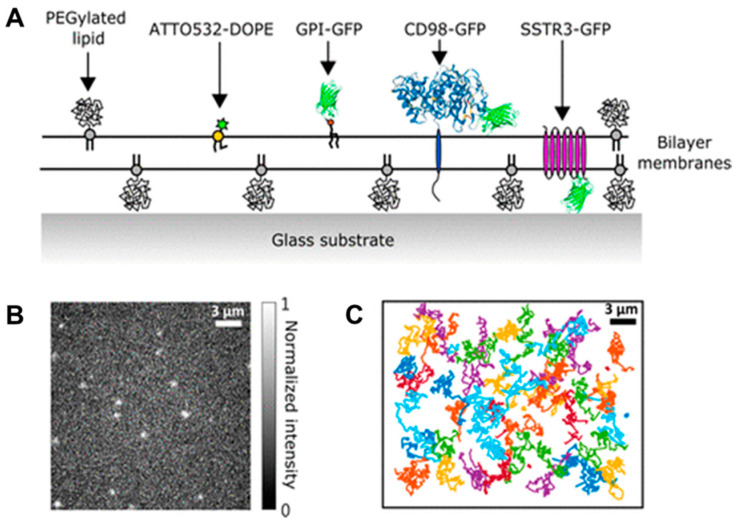
Single-molecule fluorescence imaging and tracking in polymer-cushioned plasma membrane bilayers. (**A**) Schematic of the membrane molecules studied in this work. ATTO532-DOPE is a dye-labeled phospholipid. GPI-GFP is a glycolipid, glycophosphatidylinositol (GPI), with a GFP attached to the headgroup. CD98-GFP heavy chain is a single-pass transmembrane protein, and it is fused with GFP at its C-terminus in the extracellular domain. SSTR3-GFP is a seven-pass transmembrane protein, and it is fused with GFP at its C-terminus in the cytoplasmic domain. (**B**) Snapshot of the fluorescence imaging of single GPI-GFPs. (**C**) Representative trajectories of GPI-GFPs in polymer-cushioned plasma membrane bilayers [332]. Reproduced from Wong et al. (2019) with permission (Copyright © American Chemical Society 2019).

**Figure 13 polymers-12-01003-f013:**
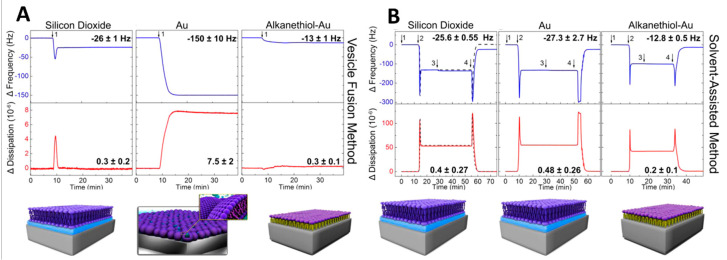
QCM-D monitoring of vesicle fusion and solvent-assisted lipid bilayer (SALB) methods on three different substrates. QCM-D frequency shift Δf (blue) and dissipation shift (ΔD, red) for the third overtone (n = 3) are measured as a function of time during lipid adsorption onto silicon dioxide, gold, and alkanethiol-coated gold. Panel (**A**) corresponds to the vesicle fusion method. DOPC lipid vesicles are added at t = 10 min (arrow 1) after establishing the baseline for the frequency and dissipation shifts. Panel (**B**) corresponds to the SALB formation method. Arrows indicate the injection of buffer (10 mM Tris, 150 mM NaCl, pH 7.5; (1)), isopropanol (2), lipid mixture (0.5 mg/mL DOPC lipid in isopropanol; (3)), and buffer exchange (4). The dashed curve in panel B represents the control experiment in which lipid is not injected. The final values of Δf and ΔD for each surface are reported. All values are given as the mean of at least three runs. The schematics show the proposed assembled lipid structures as inferred from the final frequency and dissipation shifts [352]. Reproduced from Tabaei et al. (2014) with permission (Copyright © American Chemical Society 2014).

**Figure 14 polymers-12-01003-f014:**
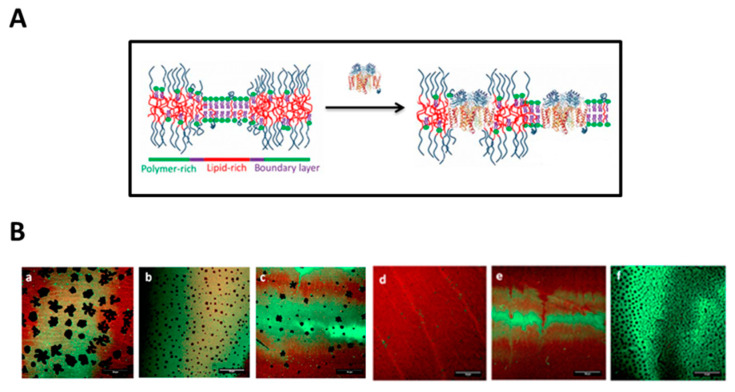
(**A**) Schematic representation of selective reconstitution of an outer membrane protein into the lipid rich domain of a polymer-lipid hybrid membrane; [371]. (**B**) CLSM micrographs showing the protein distribution in films consisting of mixtures of PDMS_65_-*b*-PMOXA_12_ and (a) DPPC (x_DPPC_ = 0.75), (b) DPPC (x_DPPC_ = 0.5), (c) DPPE (x_DPPE_ = 0.25), (d) DOPC (x_DPPC_ = 0.25), and (e) POPE (x_DPPC_ = 0.25). (f) PDMS_37_-b-PMOXA_9_ mixed with DPPE (x_DPPE_ = 0.5). Films are transferred at a surface pressure of 35 mN·m^–1^. Scale bars correspond to 50 μm [6]. Reproduced from Beales et al. (2017) (Copyright © Portland Press 2017) and Chen et al. (2010) with permission (Copyright © American Chemical Society 2010). PDMS, polydimethylsiloxane; PMOXA, poly(dimethylsiloxane); DPPC, dipalmitoylphosphatidylcholine; DPPE, 1,2-dipalmitoyl-*sn*-glycero-3-phosphoethanolamine; POPE, 1-palmitoyl-2-oleoyl-*sn*-glycero-3-phosphoethanolamine.

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
