# Peer review of "Recent Advances in Hybrid Biomimetic Polymer-Based Films: from Assembly to Applications"

_polymers, 2020, doi:10.3390/polym12051003_

Round 1
Reviewer 1 Report
Manuscript Number: polymers-765238
Title: Recent advances in hybrid biomimetic polymer-based films: from assembly to application
Polymers
This manuscript represents an effort to present a comprehensive and current review on various hybrid biomimetic membranes/films and their fabrication and applications. The review is written very clearly and with high dose of scientific knowledge and expertise. Therefore, the proposed manuscript is intended for both new and experienced scientists and researchers who want to increase their proficiency in the field of biomimetic polymer-based membranes, especially for modern applications in biotechnology, bioengineering, nanotechnology, and medicine. It is a perfect source of references serving also as a guide for practices that wish to improve and expand their knowledge on biomimetic membranes.
The paper covers the whole range of the hottest problems connected with recent developments and advancements from approaches used for synthesis of hybrid biomimetic polymer-based films. In particular, the authors discuss hybrid films based on polyelectrolytes, polymer brushes, tethers and cushions from synthetic polymers, block copolymers with combinations with macromolecules, such as lipids, proteins, enzymes, biopolymers, and nanoparticles. A special emphasis is put also on the analysis of the achieved applications including DNA analysis, drug delivery, sensors, and bioelectronics.
The paper begins with a review on assembly, properties and applications of hybrid polielectrolyte-based (such as: polyelectrolyte-natural polymers, polyelectrolyte-nucleic acids, polyelectrolyte-proteins, polyelectrolyte-enzymes) membranes. This is followed by hybrid films based on polymer brushes obtained by grafting “from” and “to” approaches. The properties and applications of such polymer brushes are discussed as well. In my opinion, this section is extremely important contribution to the scientific discussion on new functionality of such membranes and development of modern devices. However, there is a lack of discussed examples for grafting brushes on different substrates (e.g., anodic aluminum oxide) and their possible applications. In this aspect, some useful references are: M. Szuwarzynski et al., Chem. Mater., 25 (2013) 514−520; J. Jang et al., Chem. Lett., 39 (2010) 1190−1191; Q. Fu et al., J. Am. Chem. Soc., 126 (2004) 8904−8905. The next section deals with the synthesis, properties and some novel applications of polymer-lipid tethered and cushioned composite films. In the last section, a complete overview on amphiphilic block copolymers and phospholipids films (as examples of artificial membranes) is given.
The chapter is amply illustrated throughout. The drawings and images depict perfectly the main content of the chapters.
Some odd remarks:
- List of abbreviations is a must. The authors used a lot of abbreviations, and some of them are not even explained (e.g., see line 190, 209, 215, 288, 327, 348 and …) or explained further in the text but not at the place where is used (e.g., see lines: 194 and 203).
- Figure 1 is not informative for non-experts. Please describe different types of hybrid films mentioned in the figure capture.
- Line 96, typos in “hasve”.
- Line 148, missing comma after “… 53, 115]”
- Line 168. The same way of film notation should be used as in further subsections. Therefore, I suggest to use: “PE-natural polymer hybrid films”.
- Line 195, most probably excessive space before “cross-linking”.
- Line 211, missing subscripts in chemical formulas.
- Missing space after “NPs”.
- Figure 6, in the figure captions are indicated figures A-D, but there are not such labels in the image itself.
- Lines 506 and 507, missing spaces before units.
- Figure 7 caption. Uppercases are used instead of lowercases. The bacteria stains are written in italic or not (compare lines 589-590 and 747-748).
- Lines 605, 638, missing space.
- Line 607, excessive space after “Si”.
- Line 630, “Figure 8A” in bold unusually.
- Line 679, non-typical characters used in the name of TMB.
- Line 743, missing superscript in “ng/cm2”.
- Line 787, please check the reference error.
- Line 813, excessive space before “After”.
- Line 825, 882, 895 missing spaces before units.
- Line 828, excessive space before “at”.
- Line 835, missing dot at the end of sentence.
- Line 855 compare with 809, 810, and 949, different notation of the unit. Compare also with line 1143. (e.g., mW/cm or mW cm-1).
- Line 895, Excessive space before “The fastest”
- Line 898, The abbreviation “AD” is used only once in the whole text. It can be easily omitted.
- Line 915, missing space after “proteins”.
- Line 916, excessive space before “depending”.
- Line 945 compare with 961 or 965. / (slash) is used with spaces or without throughout the whole manuscript.
- Line 972, acid name written from uppercase.
- Line 995, missing space before “lipid”.
- Line 1016, unnecessary underlined dot.
- Line 1029, 1059, missing and excessive spaces.
- Line 1048, check the system of reference notation.
- Line 1135, “Figure 14” in bold unusually.
- Line 1211, excessive space.
- The reference list is carelessly composed. There are upper cases (in paper titles) instead of lower cases, journal names are differently typed (abbreviated or full names, beginning from uppercases or lowercases, in italics or not), issue number are used or omitted (compare e.g., ref. 8 and 32), missing details in some references (e.g., DOI in ref. 3, 21) and others.
To sum up, this paper might be valuable just as a general survey. Therefore, I suggest publishing the manuscript after minor revision.
Reviewer 2 Report
This review article highlights recent advances in the development and applications of biomimetic planar hybrid membranes based on four different polymer systems. Multiple approaches used for the synthesis and characterizations of these hybrid membranes are summarized along with their applications in bioengineering, biomedical and environmental areas. In general, the topic presented in this review article is interesting and exciting, and important for the development of biomimetic membranes, especially for biomimetic planar membranes. This manuscript is well organized and a pleasure to read. Therefore, I would like to recommend the acceptance of this manuscript after a minor revision.
- Given the importance of development of planar membranes, it would be great if the authors could add a few more recent examples in this area into this comprehensive review article, such as Tu et al. Nature Materials, 2020, 19, 347–354; Shen et al. Nature Communications, 2018, 9, 2294, Jin et al. Nature Communications, 2016, 7, 12252.
- Because the high stability is critical for the applications of biomimetic polymer-based films (or membranes), it would be great to add some descriptions (or a summary) in this aspect and comment on the future of developing hybrid biomimetic membranes.
- Page 5, line 179. "…due…" should be "…due to…".
- Page 5, line 203. “hyaluronic acid (HA)” should be named at the line 194 when hyaluronic acid was first mentioned in the article.
